# MASSIVELY SCALING HETEROSCEDASTIC CLASSIFIERS

**Mark Collier\*, Rodolphe Jenatton\*, Basil Mustafa, Neil Houlsby, Jesse Berent & Effrosyni Kokiopoulou**
Google AI
`{markcollier,rjenatton,basilm,neilhoulsby,jberent,kokiopou}@google.com`

## ABSTRACT

Heteroscedastic classifiers, which learn a multivariate Gaussian distribution over prediction logits, have been shown to perform well on image classification problems with hundreds to thousands of classes. However, compared to standard classifiers, they introduce extra parameters that scale linearly with the number of classes. This makes them infeasible to apply to larger-scale problems. In addition heteroscedastic classifiers introduce a critical temperature hyperparameter which must be tuned. We propose `HET-XL`, a heteroscedastic classifier whose parameter count when compared to a standard classifier scales independently of the number of classes. In our large-scale settings, we show that we can remove the need to tune the temperature hyperparameter, by directly learning it on the training data. On large image classification datasets with up to 4B images and 30k classes our method requires $14\times$ fewer additional parameters, does not require tuning the temperature on a held-out set *and* performs consistently better than the baseline heteroscedastic classifier. `HET-XL` improves ImageNet 0-shot classification in a multimodal contrastive learning setup which can be viewed as a 3.5 billion class classification problem.

## 1 INTRODUCTION

Heteroscedastic models learn an input-dependent noise term to capture uncertainty in their predictions. In deep learning, they have been used successfully in large-scale image classification (Collier et al., 2021), image segmentation (Kendall & Gal, 2017; Collier et al., 2020), regression (Lakshminarayanan et al., 2017), uncertainty quantification (Tran et al., 2022; Nado et al., 2021) and in bandit problems (Osband et al., 2021).

It is known from the economics literature that heteroscedastic classifiers are particularly suited to modelling classification problems with many classes (Train, 2009) and this has been further observed in deep learning (Collier et al., 2021). However, heteroscedastic classifiers add additional parameters to standard "deterministic" classifiers (`DET`) to define their $K \times K$ covariance matrix, with $K$ the number of classes. Even with low-rank approximations, the number of additional parameters scales linearly in $K$, thus imposing a significant cost in large-scale settings. Also, these additional parameters must be stored in long-term storage and loaded in memory which can pose problems for both storage and memory bound applications. For example, on JFT-4B, a dataset with 29,593 classes, the state-of-the-art and most scalable, to the best of our knowledge, heteroscedastic classification method `HET` (Collier et al., 2021), does not fit in memory on a large TPU slice (64 TPU v3 cells with 128 cores) when using a modest-sized ViT-L/32 base architecture.

In this paper, we propose `HET-XL` whose extra parameter count over `DET` scales independently of the number of classes. In addition, `HET` requires tuning a temperature hyperparameter $\tau$, which hinders the adoption of heteroscedastic classifiers in large-scale settings where hyperparameter sweeps are either very costly or not feasible at all. `HET-XL`, in contrast, learns $\tau$ directly on the training set. We argue and demonstrate empirically that this is feasible precisely in this very large-scale setting. Despite the improved

---

\*Equal contribution.

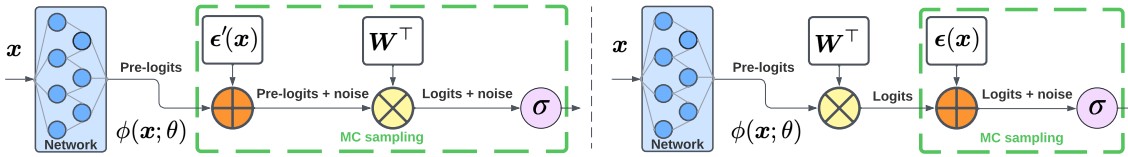

Figure 1: $\texttt{HET-XL}$ (left) injects the noise $\boldsymbol{\epsilon}'(\boldsymbol{x}) \in \mathbb{R}^D \sim \mathcal{N}(\mathbf{0}, \boldsymbol{\Sigma}'(\boldsymbol{x}))$ in the pre-logits $\boldsymbol{W}^\top(\phi(\boldsymbol{x};\theta) + \boldsymbol{\epsilon}'(\boldsymbol{x}))$, while $\texttt{HET}$ (right) adds the noise in the logits $\boldsymbol{W}^\top\phi(\boldsymbol{x};\theta) + \boldsymbol{\epsilon}(\boldsymbol{x})$ and $\boldsymbol{\epsilon}(\boldsymbol{x}) \in \mathbb{R}^K$. $\boldsymbol{\Sigma}'$ does not scale with $K$.

efficiency and ease of adoption, $\texttt{HET-XL}$ performs consistently better across large-scale image classification tasks compared to $\texttt{DET}$ and $\texttt{HET}$ and improves upon a $\texttt{DET}$ baseline in the contrastive learning setting.

**Contributions.** In summary, our contributions are:

**(1)** We develop the $\texttt{HET-XL}$ heteroscedastic classifier, whose cost of deployment is significantly reduced as compared to $\texttt{HET}$, the prior state-of-the-art. In large-scale settings, $\texttt{HET-XL}$ does not require tuning a temperature hyperparameter and has massively reduced parameter count compared to $\texttt{HET}$. Moreover, $\texttt{HET-XL}$ allows for a plug-in compatibility with existing large-scale classifiers, which is not the case for $\texttt{HET}$.
**(2)** On three image classification benchmarks—JFT 300M, ImageNet-21k and JFT-4B—and for two different popular model classes, ResNet152 and ViT-L/32, $\texttt{HET-XL}$ consistently outperforms $\texttt{HET}$ and $\texttt{HET-H}$, a new hashing-based baseline we introduce. For example, with a ResNet152 on JFT-300M, we increase precision@1 by 2.3% compared to $\texttt{HET}$, while adding about 9 times fewer parameters.
**(3)** We extend $\texttt{HET-XL}$ to contrastive learning where the method improves ImageNet 0-shot classification accuracy from 85.29% for a $\texttt{DET}$ model to 85.56% on a LiT setup (Zhai et al., 2022).

## 2 BACKGROUND ON HETEROSCEDASTIC CLASSIFIERS

We now review the core prior work that our method builds on, a wider review of related work is provided in Appendix A. We focus on classification tasks where we learn classifiers of the form

$$\texttt{softmax}(\boldsymbol{W}^\top\phi(\boldsymbol{x};\theta)) \quad \text{and} \quad \texttt{sigmoid}(\boldsymbol{W}^\top\phi(\boldsymbol{x};\theta)) \tag{1}$$

based on some training data $\mathcal{D} = \{(\boldsymbol{x}_n, y_n)\}_{n=1}^N$. A pair $(\boldsymbol{x}_n, y_n)$ corresponds to an input $\boldsymbol{x}_n$, e.g., an image, together with its label $y_n \in \{0, 1\}^K$ belonging to one, or multiple, of the $K$ classes, in the multi-class and multi-label settings, respectively. The model is parametrized by $\boldsymbol{W} \in \mathbb{R}^{D \times K}$ and the $D$-dimensional representation $\phi(\cdot;\theta)$ output by a neural network with parameters $\theta$. We have omitted the bias term to ease the presentation. Throughtout the paper, we refer to $\boldsymbol{W}^\top\phi(\boldsymbol{x};\theta) \in \mathbb{R}^K$ as the *logits*, while we will use the term *pre-logits* for $\phi(\boldsymbol{x};\theta) \in \mathbb{R}^D$. We denote the elementwise product between tensors by $\circ$.

Heteroscedastic classifiers learn an additional input-dependent noise distribution placed on the logits to capture uncertainty in the predictions of the model (Kendall & Gal, 2017; Collier et al., 2021; Train, 2009). We consider the setting where this noise is modelled by a Gaussian, leading to the class predictions

$$\mathbb{E}_{\boldsymbol{\epsilon}}\left[\sigma\left(\boldsymbol{W}^\top\phi(\boldsymbol{x};\theta) + \boldsymbol{\epsilon}(\boldsymbol{x})\right)\right] \in \mathbb{R}^K \quad \text{with} \quad \boldsymbol{\epsilon}(\boldsymbol{x}) \in \mathbb{R}^K \sim \mathcal{N}(\mathbf{0}, \boldsymbol{\Sigma}(\boldsymbol{x};\theta_{\text{cov}})), \tag{2}$$

where $\sigma$ can be either the $\texttt{softmax}$ or the $\texttt{sigmoid}$ transformation (see Fig. 1, right). Above, we have introduced the *covariance matrix* $\boldsymbol{\Sigma}(\boldsymbol{x};\theta_{\text{cov}})$ of $\boldsymbol{\epsilon}$ that is parametrized by $\theta_{\text{cov}}$; we will describe $\boldsymbol{\Sigma}$ in more details in Section 2.2. The resulting conditional probability $p(y|\boldsymbol{x}; \{\boldsymbol{W}, \theta, \theta_{\text{cov}}\})$ in Eq. (2) is used to train the model on $\mathcal{D}$ by maximum likelihood and to make predictions at evaluation time.

### 2.1 ESTIMATING THE HETEROSCEDASTIC PREDICTIONS

We will focus on the $\texttt{HET}$ method from Collier et al. (2021) as it obtains state-of-the-art performance on several benchmarks and operates at the largest scale. As shown in Eq. (2), heteroscedastic modelling requires

marginalizing over the noise $\epsilon(\boldsymbol{x})$. The corresponding expectation cannot be solved analytically (Lu et al., 2020) and Collier et al. (2021) use a Monte Carlo (MC) estimate by sampling from $\mathcal{N}(\boldsymbol{0}, \boldsymbol{\Sigma}(\boldsymbol{x}; \theta_{\text{cov}}))$, Fig. 1.

By relating Eq. (2) to a generative process where $\sigma$ approximates a discrete choice (Train, 2009), Collier et al. (2020; 2021) further consider a temperature parameter $\tau > 0$ to control this approximation. More precisely, $\sigma$ in Eq. (2) is replaced by $\sigma_\tau$ so that $\sigma_\tau(\boldsymbol{u}) = \sigma(1/\tau \cdot \boldsymbol{u})$. We present more formally the construction of the HET classifier in Appendix A.1. Importantly, Collier et al. (2020) show that $\tau$ is crucial to regulating a bias-variance trade-off between the bias with respect to the generative process and the variance of the MC estimate. $\tau$ is tuned on a held-out set and the test performance is sensitive to its choice.

## 2.2 PARAMETRIZING THE COVARIANCE MATRIX

A central component of heteroscedastic classifiers is the covariance matrix $\boldsymbol{\Sigma}(\boldsymbol{x}; \theta_{\text{cov}}) \in \mathbb{R}^{K \times K}$. It enables the model to learn which regions of the input space have noisy labels and what the correlations across classes are in those regions (Collier et al., 2021). When clear from the context, we will omit the parameters $\theta_{\text{cov}}$.

Making $\boldsymbol{\Sigma}(\boldsymbol{x})$ dependent on the inputs is challenging. Let us assume that $\boldsymbol{\Sigma}(\boldsymbol{x}) = \boldsymbol{L}(\boldsymbol{x})^\top \boldsymbol{L}(\boldsymbol{x})$ is positive definite with the full-rank matrix $\boldsymbol{L}(\boldsymbol{x}) \in \mathbb{R}^{K \times K}$ that we vectorize as $\text{vec}(\boldsymbol{L}(\boldsymbol{x})) \in \mathbb{R}^{K^2}$. Even simply using a linear transformation of the $D$-dimensional pre-logits to parameterize $\text{vec}(\boldsymbol{L}(\boldsymbol{x}))$ as

$$\text{vec}(\boldsymbol{L}(\boldsymbol{x})) = \boldsymbol{C}\phi(\boldsymbol{x}; \theta) \in \mathbb{R}^{K^2} \quad \text{with} \quad \boldsymbol{C} \in \mathbb{R}^{K^2 \times D}, \tag{3}$$

is not feasible due to the large number of elements in $\theta_{\text{cov}} = \{\boldsymbol{C}\}$ (for instance, $K = 29{,}593$ and $D = 2048$ in some of our settings in Section 6). We could restrict $\boldsymbol{\Sigma}(\boldsymbol{x})$ to be a diagonal matrix (Kendall & Gal, 2017), scaling down $\boldsymbol{C}$ to $\mathbb{R}^{K \times D}$, but this comes with a drop in performance (Collier et al., 2021). Instead, a low-rank parametrization, with $R \ll K$, of the form

$$\boldsymbol{\Sigma}(\boldsymbol{x}) = \boldsymbol{V}(\boldsymbol{x})^\top \boldsymbol{V}(\boldsymbol{x}) + \text{diag}(\boldsymbol{d}(\boldsymbol{x})) \quad \text{with} \quad \boldsymbol{V}(\boldsymbol{x}) \in \mathbb{R}^{R \times K} \quad \text{and} \quad \boldsymbol{d}(\boldsymbol{x}) \in \mathbb{R}_+^K \tag{4}$$

offers a good trade-off between memory footprint and performance of the classifier (Collier et al., 2021). In that case, using a linear transformation of the pre-logits, as above, leads to $\theta_{\text{cov}}$ of size $\mathcal{O}(DKR + DK)$. Collier et al. (2021) consider a further optimized parametrization whereby $\boldsymbol{V}(\boldsymbol{x}) = \boldsymbol{J} \circ (\boldsymbol{1}_R \boldsymbol{v}(\boldsymbol{x})^\top)$ where $\boldsymbol{J} \in \mathbb{R}^{R \times K}$ and $\boldsymbol{v}(\boldsymbol{x}) \in \mathbb{R}^K$, with $\theta_{\text{cov}}$ thus scaling in $\mathcal{O}(DK + KR)$ (in Appendix L, we discuss slightly more expressive variants of Eq. (4) that do not increase the parameter count). Despite these advances, $\mathcal{O}(DK + KR)$ remains restrictive for modern large models and problems with a large number of classes.

## 3 HET-XL: SCALING HETEROSCEDASTIC CLASSIFIERS TO VERY LARGE $K$

Heteroscedastic classifiers have resulted in impressive performance gains across many tasks, see Section 1. Empirically those improvements have been the biggest in large-scale classification tasks with many classes (Tran et al., 2022; Collier et al., 2021). However, their additional parameter count scales linearly in the number of classes $K$, see Section 2. The increase in parameter count and corresponding memory use can be large relative to the performance gains. For example, a DET ResNet152 applied to ImageNet-21k has 102.9M parameters, while a HET ResNet152 has 88% more parameters than DET. HET provides a 1.4% increase in precision@1 for this task, however the additional cost may be considered too high for this gain.

In this section we will introduce our proposed solution, the HET-XL classifier. HET-XL has significantly reduced deployment cost when $K$ is large and still provides performance gains over HET, whose large parameter count can lead to overfitting, see Section 6. On the above ImageNet-21k example, HET-XL provides a 2.4% performance gains with only 8% more parameters compared to DET.

Table 1: Summary of the properties of `HET` and `HET-XL`.

|  | Extra parameter count | Automated tuning of $\tau$ | Plug-in compatibility with existing classifiers |
|---|---|---|---|
| `HET` | $\mathcal{O}(DK + KR)$ | ✗ | ✗ |
| `HET-XL` | $\mathcal{O}(D^2 + DR)$ | ✓ | ✓ |

### 3.1  `HET-XL`

The additional parameter count of `HET` compared to `DET` scales in $\mathcal{O}(DK + KR)$ because the noise term $\epsilon$ is directly parameterized in the logit space with dimension $K$. We make the simple yet important observation: If we directly inject the noise term in the pre-logits $\phi(\boldsymbol{x}; \theta) \in \mathbb{R}^D$, and then reuse the *existing* linear transformation $\boldsymbol{W} \in \mathbb{R}^{D \times K}$ from pre-logits to logits, we can remove this dependency on the number of classes $K$. This core idea is illustrated in Fig. 1. We refer to the resulting extension of `HET` as `HET-XL`. For `HET-XL`, we thus replace Eq. (2) with

$$\mathbb{E}_{\boldsymbol{\epsilon}'} \left[ \sigma \left( \boldsymbol{W}^\top (\phi(\boldsymbol{x}; \theta) + \boldsymbol{\epsilon}'(\boldsymbol{x})) \right) \right] \quad \text{with} \quad \boldsymbol{\epsilon}'(\boldsymbol{x}) \in \mathbb{R}^D \sim \mathcal{N}(\boldsymbol{0}, \boldsymbol{\Sigma}'(\boldsymbol{x}; \theta_{\text{cov}})), \tag{5}$$

where crucially $\boldsymbol{\epsilon}' \in \mathbb{R}^D$ and the covariance matrix $\boldsymbol{\Sigma}'(\boldsymbol{x}; \theta_{\text{cov}}) \in \mathbb{R}^{D \times D}$ apply to the pre-logits. Therefore, when we parameterize $\boldsymbol{\Sigma}'$ as per the `HET` parametrizations recalled in Section 2.2, the additional parameter count of $\theta_{\text{cov}}$ compared to `DET` scales in $\mathcal{O}(D^2 + DR)$ and $\mathcal{O}(D^2 R)$, respectively. In large-scale settings $D$ is often small relative to $K$. For example, for JFT-4B, $K = 29,593$ while $D = 1024$ for a ViT-L and $D = 2048$ for a ResNet152, two representative large-scale image classification models.

Because of the properties of Gaussian distributions under linear transformations, $\boldsymbol{W}^\top(\phi(\boldsymbol{x}; \theta) + \boldsymbol{\epsilon}'(\boldsymbol{x}))$ still defines a Gaussian distribution in logit space: $\mathcal{N}(\boldsymbol{W}^\top \phi(\boldsymbol{x}; \theta), \boldsymbol{W}^\top \boldsymbol{\Sigma}'(\boldsymbol{x}) \boldsymbol{W})$. However, the covariance matrix, or some decomposition thereof, is never explicitly computed in this space. Interestingly, in Table 15 (Appendix I), we show that the choice of sharing the $\boldsymbol{W}$ transformation between the pre-logits and the noise samples does not sacrifice performance compared to separate transformations $\boldsymbol{W}^\top \phi(\boldsymbol{x}; \theta) + (\boldsymbol{W}')^\top \boldsymbol{\epsilon}'(\boldsymbol{x})$.

**Plug-in compatibility with existing large-scale classifiers.** In extreme classification tasks, where $K$ may be in the millions, the standard matrix-vector multiplication $\boldsymbol{W}^\top \phi(\boldsymbol{x}; \theta)$ may have to be replaced by a more scalable logic, involving for instance a distributed lookup of the active classes (Zhang et al., 2018; Song et al., 2020). Let us assume this logic is abstracted by an API of the form `get_logits`$(\phi(\boldsymbol{x}; \theta))$. `HET-XL` can directly piggyback that functionality by using `get_logits`$(\phi(\boldsymbol{x}; \theta) + \boldsymbol{\epsilon}'(\boldsymbol{x}))$ while `HET` cannot be easily adapted to this case. Table 1 summarises the differentiating properties of `HET` and `HET-XL`.

## 4  LEARNING THE TEMPERATURE $\tau$

As described in Section 2.1, the temperature $\tau$ of `HET` controls a bias-variance trade-off. The resulting performance is sensitive to the value of $\tau$ whose choice is often dataset-dependent (Collier et al., 2020; 2021). In practice, $\tau$ is included in the hyperparameter sweep which may become prohibitively expensive to run at scales considered here. At the same time, bypassing this step or making the sweep too coarse may degrade performance (Collier et al., 2020). We thus investigate strategies to automatically tune $\tau$.

### 4.1  A REPRESENTATIVE BENCHMARK

As a working study, we consider the tuning of $\tau$ in `HET-XL` for a ResNet50x2 model on ImageNet-21k trained for 7 epochs (the dataset will be presented in more details in Section 6). "x2" refers to the factor multiplying the width of each layer (Szegedy et al., 2017). This setting is representative of the large-scale tasks we deal with in our subsequent experiments, although slightly scaled down, both in terms of model size and training regime. We do so to be able to compare with a wider spectrum of methods and assess the

Table 2: Comparisons (means $\pm$ standard errors) of methods to tune $\tau$ for a ResNet50x2 on ImageNet-21k.

| Method | $\downarrow$NLL | $\uparrow$Prec@1 | Single training? | No validation set needed? |
|--------|------|--------|------------------|---------------------------|
| Grid search | $6.08_{\pm 0.04}$ | $0.433_{\pm 0.002}$ | ✗ | ✗ |
| Bayesian optimization | $6.03_{\pm 0.06}$ | $0.428_{\pm 0.000}$ | ✗ | ✗ |
| Luketina et al. (2016) | $6.08_{\pm 0.04}$ | $0.425_{\pm 0.000}$ | ✓ | ✗ |
| Simple training (as in `HET-XL`) | $5.99_{\pm 0.04}$ | $0.435_{\pm 0.001}$ | ✓ | ✓ |

validity of the results with multiple replications (we will report the average and the standard error over three replications). We provide further details of the benchmark in Appendix G.

**Approaches with multiple trainings.** To obtain gold-standard references, we consider methods that tune $\tau$ based on multiple successive trainings. In particular, we focus on grid search (GS) (assuming a grid of values for $\tau$ informed by an experienced user; see Appendix G) and Bayesian optimisation (BO) (Snoek et al., 2012) which we run for 30 full trainings; we use the implementation of Song et al. (2022); Golovin et al. (2017). None of those approaches would scale up to our largest experiments in Section 6.

**Online gradient-based tuning.** Given that $\tau$ is a one-dimensional continuous hyperparameter, it is meaningful to consider approaches that optimize a validation objective by gradient descent, typically following a bi-level formulation (Maclaurin et al., 2015; Luketina et al., 2016; Pedregosa, 2016; Franceschi et al., 2018; Lorraine et al., 2020). Those approaches need to approximate costly high-order derivatives that account for the dependency of the hyperparameters in the validation objective. In our study, we will take Luketina et al. (2016) as a representative gradient-based method that considers the hyperparameter dependency only through the current training step gradient (we formally describe their approach in Appendix G). Moreover, because of the particular structure of $\tau$—explicitly appearing both at training and validation time, unlike optimisation-related or regularisation-related hyperparameters that surface only at training time—it is feasible to consider and evaluate a simpler alternative gradient estimator, see Appendix G.

**Do we need a validation set? Simple training of $\tau$.** The experiments we consider in Section 6 bring into play datasets with a large number of data points (from 12.8M up to 4B) over which training is limited to only a few epochs. As a result, it is natural to wonder to what extent metrics collected on the training set are faithful to metrics computed on a proper held-out validation set, since overfitting seems unlikely to happen. We thus consider an approach wherein $\tau$ is treated as a simple training variable, optimized via the training objective like $\{W, \theta, \theta_{\text{cov}}\}$. In the contrastive learning setting, Jia et al. (2021); Radford et al. (2021); Mustafa et al. (2022) already successfully learned a temperature during training, although in a different context where $\tau$ was not responsible for a specific bias-variance trade-off like in `HET-XL`.

## 4.2 RESULTS AND DISCUSSION

We summarise the results in Table 2. The main conclusion is that *there is no opportunity cost in using the simplest approach*—tuning $\tau$ as a standard training parameter—even when compared to more advanced, e.g., Luketina et al. (2016), and more expensive methods like GS and BO requiring multiple trainings. Remarkably, we still observe the benefit of "Simple training" in our larger-scale experiments, see Table 8.

In the perspective of having a plug-in deployment of `HET-XL`, "Simple training" has the advantage over Luketina et al. (2016) not to require an extra validation set to iterate over during training, which may be constraining in production pipelines. In Appendix G, we try to replace the validation set of Luketina et al. (2016) by some held-out subset, say 25%, of the (training) batches, but we observe some drop in performance (past the first epoch, this held-out subset of the batches does not behave as actual held-out data anymore).

Let us consider the split of the entire training set $\mathcal{D}$ into $\mathcal{D}_{\text{train}} \cup \mathcal{D}_{\text{val}}$. We could hypothesise that GS and BO perform worse because their corresponding models are trained on less data, i.e., just $\mathcal{D}_{\text{train}}$ (with $\mathcal{D}_{\text{val}}$ being

used to guide the tuning), as opposed to $\mathcal{D}_{\text{train}} \cup \mathcal{D}_{\text{val}}$ for "Simple training". In Appendix G, we check the effect of retraining on $\mathcal{D}_{\text{train}} \cup \mathcal{D}_{\text{val}}$ the best models obtained by GS and BO, with no change in our conclusion.

Hence, from now on, we assume that the temperature $\tau$ in `HET-XL` is trained like any other model parameter.

## 5 CONTRASTIVE LEARNING

Contrastive learning is a growing area of research (Radford et al., 2021; Jia et al., 2021; Zhai et al., 2022; Yu et al., 2022; Mustafa et al., 2022) where pairs of data, possibly of different modalities, e.g., image and text, are learned to be aligned. Contrastive models have shown strong performance in zero-shot image classification and retrieval, as well as when learning representations for downstream tasks.

### 5.1 HETEROSCEDASTIC CONTRASTIVE LEARNING

**Contrastive learning as classification.** We can view contrastive learning as a massive classification problem. Assume a dataset of $N$ image-text pairs $\mathcal{D} = \{(\boldsymbol{a}_n, \boldsymbol{b}_n)\}_{n=1}^N$. We are trying to learn well-aligned representations of $(\boldsymbol{a}_n, \boldsymbol{b}_n)$ that we denote by $(\boldsymbol{\alpha}_n, \boldsymbol{\beta}_n)$, with $\boldsymbol{\alpha}_n, \boldsymbol{\beta}_n \in \mathbb{R}^D$ output by some neural network.

We want to push $(\boldsymbol{\alpha}_n, \boldsymbol{\beta}_n)$ to be similar while $\{(\boldsymbol{\alpha}_i, \boldsymbol{\beta}_n)\}_{i \neq n}$ are encouraged to be dissimilar. In this sense, we can see this setting as having two classification tasks, one from *image to text*—given $\boldsymbol{\alpha}_n$, identify the most similar $\boldsymbol{\beta}_n$—and one from *text to image*–given $\boldsymbol{\beta}_n$, identify the most similar $\boldsymbol{\alpha}_n$—with $K = N$ classes in both cases. Since $N \gg 1$, those two "global" classification tasks are approximated by "local", batch-level classification tasks. Indeed, during training, given a batch $\mathcal{B} \subset \{1, \ldots, N\}$ of size $B = |\mathcal{B}| \ll N$, we focus on the contrastive loss $\mathcal{L}(\mathcal{B}) = \frac{1}{2B} \sum_{n \in \mathcal{B}} \{\mathcal{L}_n^{\text{image-text}}(\mathcal{B}) + \mathcal{L}_n^{\text{text-image}}(\mathcal{B})\}$ where we have defined

$$\mathcal{L}_n^{\text{image-text}}(\mathcal{B}) = -\log\left\{\frac{e^{\boldsymbol{\alpha}_n^\top \boldsymbol{\beta}_n}}{\sum_{i \in \mathcal{B}} e^{\boldsymbol{\alpha}_n^\top \boldsymbol{\beta}_i}}\right\} \quad \text{and} \quad \mathcal{L}_n^{\text{text-image}}(\mathcal{B}) = -\log\left\{\frac{e^{\boldsymbol{\alpha}_n^\top \boldsymbol{\beta}_n}}{\sum_{i \in \mathcal{B}} e^{\boldsymbol{\alpha}_i^\top \boldsymbol{\beta}_n}}\right\}. \tag{6}$$

**Heteroscedastic classification for contrastive learning.** Contrastive learning datasets have been primarily constructed automatically by web scraping (Radford et al., 2021; Jia et al., 2021; Zhai et al., 2022; Yu et al., 2022). This process leads to noisy image-text pairs (Jia et al., 2021). This noise and the above view of contrastive learning as a massive classification task suggests naturally the use of heteroscedastic models. A naive application of Eq. (2) would define two heteroscedastic classifiers with two covariance matrices

$$\boldsymbol{\Sigma}^{\text{image-text}}(\boldsymbol{a}) \in \mathbb{R}^{N \times N} \quad \text{and} \quad \boldsymbol{\Sigma}^{\text{text-image}}(\boldsymbol{b}) \in \mathbb{R}^{N \times N} \quad (\text{remember } N = K) \tag{7}$$

and for each batch $\mathcal{B}$, the appropriate sub-matrices (with rows and columns indexed by $\mathcal{B}$) would be used for the sampling of $\boldsymbol{\epsilon}$, i.e., $\boldsymbol{\Sigma}_{\mathcal{B}}^{\text{image-text}}(\boldsymbol{a}) \in \mathbb{R}^{B \times B}$ and $\boldsymbol{\Sigma}_{\mathcal{B}}^{\text{text-image}}(\boldsymbol{b}) \in \mathbb{R}^{B \times B}$. Unfortunately, the scale of $\boldsymbol{\Sigma}^{\text{image-text}}(\boldsymbol{a})$ and $\boldsymbol{\Sigma}^{\text{text-image}}(\boldsymbol{b})$ is not manageable, e.g., $N = 4\text{B}$ for some standard contrastive learning datasets (Zhai et al., 2022). We next develop a `HET-XL` style contrastive learner where the noise is injected in the representation space (i.e., $\boldsymbol{\alpha}_n$ and $\boldsymbol{\beta}_n$), enabling a scaling independent of the number of classes $K = N$.

### 5.2 CONTRASTIVE LEARNING WITH `HET-XL`

Let us focus just on text-to-image classification, as the image-to-text case is symmetric. Looking at Eq. (6), it is natural to think of the vector $\boldsymbol{\alpha}_n \in \mathbb{R}^D$ as being equivalent to the $n$-th column of $\boldsymbol{W} \in \mathbb{R}^{D \times K}$ (remembering again that $K = N$) in a standard classifier like Eq. (1). Under this viewpoint, the image network producing the representations $\{\boldsymbol{\alpha}_n\}_{n=1}^N$ acts like a lookup function into the columns of $\boldsymbol{W}$. This perspective is especially natural in the case where the image tower is fixed and not learned (Zhai et al., 2022).

In summary, we can thus mirror the standard classification setup as follows. Given a batch $\mathcal{B}$ of size $B$, we can collect the vectors $\{\boldsymbol{\alpha}_n\}_{n \in \mathcal{B}}$ into a matrix $\boldsymbol{A}_{\mathcal{B}} \in \mathbb{R}^{D \times B}$, leading to the following (homoscedastic)

Table 3: Method comparison on 3 datasets, JFT-300M trained for 7 epochs (top), ImageNet-21k trained for 90 epochs (middle) and JFT-4B trained for 1 epoch (bottom). Two architectures are used, a ResNet152x1 (left) and a ViT-L/32 (right). NLL stands for negative log-likelihood. For HET, $\tau^*$ is found by grid search.

| | | ResNet152x1 | | | ViT-L/32 | | |
|---|---|---|---|---|---|---|---|
| | **Method** | $\downarrow$**NLL** | $\uparrow$**Prec@1** | $\downarrow$**#Params** | $\downarrow$**NLL** | $\uparrow$**Prec@1** | $\downarrow$**#Params** |
| JFT-300M | DET | 8.57 | 0.429 | 95.6M | 7.83 | 0.468 | 325.3M |
| | HET $\tau = 1$ | 8.39 | 0.440 | 171.5M | 7.79 | 0.469 | 363.7M |
| | HET $\tau^*$ | 8.45 | 0.444 | 171.5M | 7.89 | 0.484 | 363.7M |
| | HET-H | 8.38 | 0.452 | 104.1M | 7.68 | 0.493 | 327.5M |
| | HET-XL | **8.11** | **0.467** | 104.1M | **7.65** | **0.498** | 327.5M |
| ImageNet-21k | DET | 5.79 | 0.461 | 102.9M | 5.72 | 0.472 | 328.9M |
| | HET $\tau = 1$ | 5.73 | 0.469 | 193.5M | 5.89 | 0.468 | 374.8M |
| | HET $\tau^*$ | 5.72 | 0.475 | 193.5M | 5.80 | 0.471 | 374.8M |
| | HET-H | 5.73 | 0.469 | 111.4M | **5.72** | **0.474** | 331.1M |
| | HET-XL | **5.71** | **0.486** | 111.4M | 5.78 | **0.474** | 331.1M |
| JFT-4B | DET | 5.44 | 0.508 | 118.8M | 5.17 | 0.539 | 336.9M |
| | HET $\tau^*$ | 5.38 | 0.512 | 241.6M | 5.19 | 0.533 | 399.1M |
| | HET-XL | **5.34** | **0.517** | 127.2M | **5.10** | **0.544** | 339M |

classifier for the input text $\boldsymbol{b}_n$, $p(\boldsymbol{a}|\boldsymbol{b}_n) = \texttt{softmax}\left(\boldsymbol{A}_{\mathcal{B}}^\top \boldsymbol{\beta}_n\right)$, echoing the form of Eq. (1). We can now extend this reasoning to apply HET-XL to contrastive learning

$$p(\boldsymbol{a}|\boldsymbol{b}_n) = \texttt{softmax}\left(\boldsymbol{A}_{\mathcal{B}}^\top (\boldsymbol{\beta}_n + \boldsymbol{\epsilon}(\boldsymbol{b}_n))\right) \quad \text{with} \quad \boldsymbol{\epsilon}(\boldsymbol{b}_n) \in \mathbb{R}^D \sim \mathcal{N}(\boldsymbol{0}, \boldsymbol{\Sigma}^{\text{text-image}}(\boldsymbol{b}_n)), \qquad (8)$$

where, as desired, $\boldsymbol{\Sigma}^{\text{text-image}}(\boldsymbol{b}_n)$ remains of size $\mathbb{R}^{D \times D}$ independent of $K$. Table 6 Appendix A.2 shows the zero-shot results. HET-XL improves ImageNet zero-shot performance from 85.29% to 85.56%.

# 6 RESULTS

## 6.1 A COMPETITIVE BASELINE HET-H: A HASHING APPROACH

The description of HET-XL (Section 3.1) has highlighted that we can improve HET by efficiently mapping a covariance defined in a lower-dimensional space to the logit space. Equipped with this insight, and to stress test our approach as competitively as possible, we design a baseline that retains the rich heteroscedastic modeling of HET without making the parameters $\theta_{\text{cov}}$ scale with respect to $K$.

To this end, we take inspiration from the "hashing trick" (Weinberger et al., 2009; Chen et al., 2015; Eban et al., 2020), notably popularised by its application to large-scale advertising problems (Chapelle et al., 2014). The interested readers can find more background about the hashing trick in Appendix A.3. Let us consider a maximum covariance dimension $H$, with $H \ll K$. Moreover, analogously to the "hashing trick", let $h$ be a hashing function going from $\{1, \dots, K\}$ to $\{1, \dots, H\}$. Representing the hashing map via the matrix $\boldsymbol{H} \in \{0,1\}^{H \times K}$ with the $(j,k)$-th entry being non-zero when $h(k) = j$, we obtain

$$\mathbb{E}_{\boldsymbol{\epsilon}''}\left[\sigma\left(\boldsymbol{W}^\top \phi(\boldsymbol{x};\theta) + \boldsymbol{H}^\top \boldsymbol{\epsilon}''(\boldsymbol{x})\right)\right] \quad \text{with} \quad \boldsymbol{\epsilon}''(\boldsymbol{x}) \in \mathbb{R}^H \sim \mathcal{N}(\boldsymbol{0}, \boldsymbol{\Sigma}''(\boldsymbol{x};\theta_{\text{cov}})), \qquad (9)$$

where, like in Section 3.1, the covariance $\boldsymbol{\Sigma}''(\boldsymbol{x};\theta_{\text{cov}}) \in \mathbb{R}^{H \times H}$ is in a lower-dimensional space. The mapping induced by the hashing function $h$ can lead to collisions, i.e., some different classes will be modeled by the same entry in the covariance matrix. In advertising problems, the collisions in features have a regularizing effect (Chapelle et al., 2014). Since $\boldsymbol{H}$ is defined via the hashing function $h$, we only manipulate $\boldsymbol{H}$

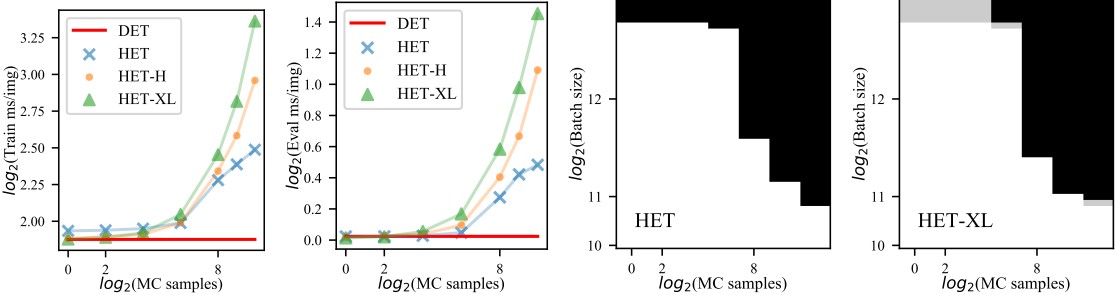

Figure 2: ImageNet-21k ResNet152x1 latency (left) and memory usage (right). Black areas mean that the method goes out of memory for a given number of MC samples and batch size (the maximum batch size is 6,144 and maximum MC samples is 1,000). Grey areas mean that `HET-XL` fits in memory but `HET` does not. `HET-XL` has lower memory usage than `HET`. The relative latency between the methods is sensitive to the number of MC samples. At lower MC samples, `HET-XL` is faster than `HET` and vice versa. It is possible to reduce `HET-XL`'s MC samples with minimal effect on predictive performance, see Appendix H.

through simple indexing operations. To make a fair comparison with `HET-XL`, we will take $H = D$, noting that if $\boldsymbol{H} = \boldsymbol{W}$, we fall back to `HET-XL`. We name this method `HET-H` for heteroscedastic hashing.

## 6.2 LARGE-SCALE IMAGE CLASSIFICATION

We evaluate `HET-XL` on three image classification benchmarks: (i) Imagenet-21k, which is an expanded version of the ILSVRC2012 ImageNet benchmark (Deng et al., 2009; Beyer et al., 2020) with 12.8M training examples and 21,843 classes, (ii) JFT-300M (Hinton et al., 2015; Chollet, 2017; Sun et al., 2017; Kolesnikov et al., 2019) with over 300M training images and 18,291 classes and (iii) JFT-4B, an expanded version of the JFT dataset with over 4B training examples and 29,593 classes. All these datasets have previously been used to benchmark the `HET` method (Collier et al., 2021; Tran et al., 2022).

We evaluate on two performant large-scale image classification architectures; ResNet152 (Szegedy et al., 2017) and ViT-L/32 (Dosovitskiy et al., 2020). Further experimental details and hyperparameters are given in Appendix N. We build upon open-source implementations (Nado et al., 2021; Tran et al., 2022). The code to implement `HET-XL` as a drop-in classifier last-layer, and the scripts to replicate our ImageNet-21k results are publicly available on GitHub (`https://github.com/google/uncertainty-baselines`). The JFT and contrastive learning code is not publicly released as it makes use of proprietary datasets.

**Analysis and discussion.** For both architectures on JFT-300M and ImageNet-21k, there is a consistent ranking of the methods, see Table 3. `HET` outperforms `DET`. `HET-H` outperforms `HET`, with one exception (ImageNet-21k ResNet152) and `HET-XL` outperforms or is on par with all other baselines.

It is also important to quantify the various costs of each method. Each table shows the parameter count of the different methods. Of course, the `DET` baseline always has the lowest parameter count. We also choose the $H$ hyperparameter of `HET-H` to equalize the number of parameters with the `HET-XL` method. These two methods have significantly reduced parameter count compared to the `HET` method. This is an important consideration for real-world deployment which often has tight storage constraints (Song et al., 2020).

Parameters must be loaded into memory, impacting memory usage. While it is more difficult to quantify memory usage and latency considering interactions with XLA compilation (Sabne, 2020), we follow the best practices from the literature (Zhai et al., 2022). To measure memory usage, we evaluate which models

Table 4: Effect of data scaling. ViT-L/32 JFT 300M comparison of methods. See Table 3 for 7 epoch results.

|  | 14 epochs | | 28 epochs | |
|---|---|---|---|---|
| **Method** | **↓NLL** | **↑Prec@1** | **↓NLL** | **↑Prec@1** |
| DET | 7.61 | 0.487 | 7.51 | 0.496 |
| HET | 7.62 | 0.484 | 7.41 | 0.505 |
| HET-XL | **7.41** | **0.518** | **7.33** | **0.523** |

Table 5: Effect of model scaling: ResNet152x2 on JFT-300M. See Table 3 for ResNet152x1 results.

| **Method** | **↓NLL** | **↑Prec@1** |
|---|---|---|
| DET | 7.83 | 0.490 |
| HET-XL | **7.42** | **0.526** |

fit into memory at varying batch sizes for a series of MC samples. As it is demonstrated in Fig. 2 and Fig. 5 (Appendix F), HET-XL has lower memory usage than HET and slightly lower memory usage to HET-H.

The conclusions about HET-XL's latency are more mixed. The relative latency of the methods depends on the number of MC samples. There is a trade-off between the cost of computing the samples from the $\Sigma(x)$ matrix for HET and the cost of the $W$ matrix multiplication in HET-XL. For smaller numbers of MC samples, HET-XL is faster than HET and vice versa. We provide a detailed analysis of this phenomenon in Appendix F.1. When latency is a concern, the number of MC samples in the HET-XL method can be significantly reduced with only a small impact on predictive performance, see Table 14 (Appendix H) for a sensitivity analysis. In this regime, HET-XL is faster, uses less memory, has substantially reduced parameter count and still provides significant performance gains over HET.

**Model and data scaling ablations.** In Table 5, we see the results when we increase the model scale from a ResNet152x1 to a ResNet152x2. While both methods benefit from model scaling, the relative gains between the DET and HET-XL methods remain roughly the same. Similarly, we scale the number of training epochs for JFT-300M Table 4 and JFT-4B Table 10 (Appendix D). Again all methods benefit from more training examples and the relative differences between methods remain roughly the same.

**Learning $\tau$ ablation.** In Table 8 (Appendix B), we run an ablation to separate the effect of learning $\tau$ from other features of the HET-XL method. We see that learning $\tau$ accounts for part but not all of the gains over HET. For example when we do not learn $\tau$, HET-XL on JFT-300M still outperforms HET by 1.6% in precision@1, further increasing to 2.3% by learning $\tau$. Not only does removing the temperature hyperparameter ease the adoption of HET-XL, the performance gains also empirically confirm the insight that we can learn $\tau$ in the large-scale setting (see Section 4). Outside of that large-scale setting, i.e., where models are more prone to overfitting by training for further epochs, we demonstrate that learning $\tau$ becomes indeed less useful, see Table 9 in Appendix B. How precisely we parameterize $\tau$ and the HET-XL's insensitivity to choices in this parameterization is set out in Appendix K.

# 7 CONCLUSIONS AND FUTURE WORK

We have developed HET-XL that consistently outperforms the DET and HET approaches in image classification and zero-shot classification from a contrastively trained model. In our large-scale settings, not only does HET-XL substantially reduce the parameter count compared to HET, it also removes the need to tune the extra temperature hyperparameter on a held-out set. As a result, HET-XL significantly eases the deployment of heteroscedastic classifiers in problems with many classes. In future research, we would like to study approaches that can bypass Monte Carlo sampling, which can negatively impact the latency and memory of HET-XL, see Figure 2. For example, we could exploit approaches based on some Taylor expansion of Eq. (2) (Collier et al., 2021) or based on mean-field approximations (Lu et al., 2020). One central question is how the simultaneous increase in bias introduced by those approximations together with a reduction in variance in the estimation of the integral of Eq. (2) affect the training and end performance of the classifiers.

**Acknowledgements.** We thanks Lucas Beyer for his help with the FLOPs, latency and memory usage measurements.

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

## A    RELATED WORK

Our methodology connects with several lines of research, as described in the following paragraphs.

**Uncertainty quantification.**    Total uncertainty can be divided into epistemic/model uncertainty and aleatoric/data uncertainty (Kendall & Gal, 2017). The neural network uncertainty quantification literature has primarily focused on epistemic uncertainty (Blundell et al., 2015; Gal & Ghahramani, 2016; Lakshminarayanan et al., 2017; Maddox et al., 2019). Aleatoric uncertainty has the interesting property that it is irreducible, meaning that even in the large-scale settings such as those considered in this paper there may be considerable aleatoric uncertainty. Heteroscedastic models are a popular method to model aleatoric uncertainty in regression (Bishop & Quazaz, 1997; Williams & Rasmussen, 2006), classification (Kendall & Gal, 2017; Collier et al., 2020; 2021; Osband et al., 2021; Fortuin et al., 2022; Tran et al., 2022) and image segmentation (Monteiro et al., 2020). Prior work on heteroscedastic classification has not been scaled up to problems of the size of JFT-4B and would require restricting the covariance matrix parameterization, for example to the diagonal, to enable scaling. Additionally prior work, either ignores the temperature parameter leading to reduced predictive performance or requires an expensive grid search over the temperature.

**Gaussian-sigmoid and Gaussian-softmax integrals.**    A central component of our methodology lies in the computation of either Gaussian-sigmoid or Gaussian-softmax integrals (Daunizeau, 2017). In both cases, no analytical forms exist for those integrals and we need to resort to approximations. Those integrals play a central role in Bayesian inference (Bishop, 2006) and probabilistic machine learning (Murphy, 2012). For heteroscedastic classifiers, Kendall & Gal (2017); Collier et al. (2020; 2021) consider an estimation based on Monte Carlo sampling, which is feasible and practical for (say, with several hundreds to thousands samples) since the operation takes place at the final layer, requiring only a single forward pass. In a different context, for the modelling of ensembles from a single model, Lu et al. (2020) propose to use mean-field approximations that they benchmark against numerical approaches such as the unscented Kalman Filter (Wan & Van Der Merwe, 2000). Variational bounds have also been proposed in the context of variational inference, see Bouchard (2007) and references therein. Daunizeau (2017); Collier et al. (2021) further consider approximations resulting from a Taylor expansion.

To the best of our knowledge, only the approaches based on Monte-Carlo sampling and Taylor expansion have been successfully exploited for heteroscedastic classifiers where the large-scale covariance is learned during training, unlike, e.g., Lu et al. (2020).

**Extreme classification.**    Previous work has looked into classification problems with millions of classes (often referred to as extreme classification or massive classification). Zhang et al. (2018) found empirically that the class probabilities concentrate on a few classes only, which are called "active classes". Motivated by this observation the authors propose an efficient hashing-based implementation of "selective softmax", which omits the non-active classes from the softmax computation. Yuan et al. (2020) proposed an efficient training framework to handle extreme classification tasks based on random projections. The idea consists in training first a random projected softmax classifier and then recover the original classifier. Alonog the lines of approximating full softmax, Song et al. (2020) introduce a large-scale training system for 100M classification at Alibaba. The proposed approach for accurately approximating the full softmax is pretty involved, including building a $k$-NN graph on $W$ for fast retrieval of active classes, a specialized communication strategy (including gradient sparsification) and a fast continuous convergence strategy for the end-to-end system. Finally, there exist also frequency-based methods (see e.g., Grave et al. (2017) and references therein) assuming that the class distribution is imbalanced and focusing on the most frequently occurring classes. Although this may be the case in language modeling context, these methods are not suitable for problems where all classes are important (e.g., in face recognition). We emphasize that HET-XL

can be easily combined with the methods above, unlike HET, which requires custom solutions to enable the matrices defining $\mathbf{\Sigma}(\boldsymbol{x})$ to fit in memory.

## A.1 More formal background about HET

In this section, we derive the form of the HET classifier, in particular, justifying why we introduce a temperature parameter and why we use MC sampling. Our exposition follows that of Collier et al. (2020; 2021), which we reproduce here for completeness.

Following the generative process typically employed in the econometric (Train, 2009), Gaussian-process (Williams & Rasmussen, 2006) and noisy-label (Kendall & Gal, 2017) literature, we consider a utility vector $\boldsymbol{u}(\boldsymbol{x}) \in \mathbb{R}^K$ and (zero-mean) stochastic term $\boldsymbol{\epsilon}(\boldsymbol{x}) \in \mathbb{R}^K \sim \pi(\boldsymbol{x})$ so that the probability $p(c|\boldsymbol{x})$ for the class $c \in \{1, \ldots, K\}$ is defined via

$$p(c|\boldsymbol{x}) = \mathbb{E}_{\boldsymbol{\epsilon}(\boldsymbol{x}) \sim \pi(\boldsymbol{x})} \left[ \mathbb{1}\left\{ c = \underset{j \in \{1, \ldots, K\}}{\arg\max} \; \boldsymbol{e}_j^\top (\boldsymbol{u}(\boldsymbol{x}) + \boldsymbol{\epsilon}(\boldsymbol{x})) \right\} \right]$$

with $\boldsymbol{e}_j \in \{0, 1\}^K$ denoting the one-hot vector with a one at the $j$-th entry and zeroes elsewhere. In words, the probability for $c$ corresponds to how often the utility, in expectation with respect to the stochastic perturbation, is maximal for $c$.

A classical, input-independent instantiation for $\pi(\boldsymbol{x}) = \pi$ is an i.i.d. Gumbel distribution, thus recovering the standard softmax with logits given by $\boldsymbol{u}(\boldsymbol{x})$ (Train, 2009). HET considers instead $\pi(\boldsymbol{x}) = \mathcal{N}(\mathbf{0}, \mathbf{\Sigma}(\boldsymbol{x}))$, in particular extending the "identical" and "independent" properties of the Gumbel distribution. We have

$$
\begin{aligned}
p_{\text{HET}}(c|\boldsymbol{x}) &= \mathbb{E}_{\boldsymbol{\epsilon}(\boldsymbol{x}) \sim \mathcal{N}(\mathbf{0}, \mathbf{\Sigma}(\boldsymbol{x}))} \left[ \mathbb{1}\left\{ c = \underset{j \in \{1, \ldots, K\}}{\arg\max} \; \boldsymbol{e}_j^\top (\boldsymbol{u}(\boldsymbol{x}) + \boldsymbol{\epsilon}(\boldsymbol{x})) \right\} \right] \\
&= \mathbb{E}_{\boldsymbol{\epsilon}(\boldsymbol{x}) \sim \mathcal{N}(\mathbf{0}, \mathbf{\Sigma}(\boldsymbol{x}))} \left[ \lim_{\tau \to 0} \boldsymbol{e}_c^\top \, \text{softmax}\left( \frac{1}{\tau}(\boldsymbol{u}(\boldsymbol{x}) + \boldsymbol{\epsilon}(\boldsymbol{x})) \right) \right] \\
&\approx \mathbb{E}_{\boldsymbol{\epsilon}(\boldsymbol{x}) \sim \mathcal{N}(\mathbf{0}, \mathbf{\Sigma}(\boldsymbol{x}))} \left[ \boldsymbol{e}_c^\top \, \text{softmax}\left( \frac{1}{\tau}(\boldsymbol{u}(\boldsymbol{x}) + \boldsymbol{\epsilon}(\boldsymbol{x})) \right) \right] \quad \text{for some small } \tau > 0 \\
&\approx \frac{1}{S} \sum_{s=1}^{S} \boldsymbol{e}_c^\top \, \text{softmax}\left( \frac{1}{\tau}(\boldsymbol{u}(\boldsymbol{x}) + \boldsymbol{\epsilon}^s(\boldsymbol{x})) \right) \quad \text{for } S \text{ samples } \boldsymbol{\epsilon}^s(\boldsymbol{x}) \sim \mathcal{N}(\mathbf{0}, \mathbf{\Sigma}(\boldsymbol{x})).
\end{aligned}
$$

The last two lines show how the temperature controls the approximation of the actual $\arg\max$ while the Monte Carlo sampling approximates the softmax-Gaussian integral with $S$ samples.

## A.2 Contrastive learning

We evaluate HET-XL added to a LiT style contrastive learner (Zhai et al., 2022). During training, HET-XL takes just one MC sample. At test time, the deterministic logits (using no MC samples) are used for zero-shot

Table 6: HET-XL LiT vs. DET LiT zero-shot accuracy $\pm 1$ std. error on ImageNet, CIFAR-100 and Oxford-IIIT Pets datasets. Average over 3 random seeds. [†]$p < 0.01$ in a two-sample unequal variance t-test.

| Method | ↑ImageNet | ↑CIFAR-100 | ↑Oxford-IIIT Pet |
|---|---|---|---|
| DET | 85.29[†] ($\pm$0.0242) | 83.39 ($\pm$0.147) | 97.87 ($\pm$0.072) |
| HET-XL | 85.56[†] ($\pm$0.027) | 83.41 ($\pm$0.101) | 97.75 ($\pm$0.071) |

Table 7: Comparison of different placements of the normalization operation in contrastive learning with `HET-XL`. ImageNet/CIFAR-100/Oxford-IIIT Pet zero-shot accuracies are shown.

| Method | Post-noise normalization=False | Post-noise normalization=True |
|---|---|---|
| **Pre-noise normalization=False** | 82.22/81.48/95.09 | 84.07/82.55/95.50 |
| **Pre-noise normalization=True** | 84.37/**82.81**/97.44 | **84.65**/82.56/**97.63** |

classification. This test procedure combined with the single MC sample during training means the cost of adding a `HET-XL` head is negligible compared to the overall network. In Appendix M, we run an ablation which justifies this choice by demonstrating no change in the zero-shot accuracy of a `HET-XL` contrastive learner when using MC sampling versus a deterministic representation at test time.

For our experiments, we use a ViT-g text encoder network and ViT-G image encoder (Dosovitskiy et al., 2020). The setup is trained on 256 v3 TPU cells (512 cores) for 18 billion examples with a batch size of 16,384, on the same data as the original LiT model. Further experimental details are given in Appendix N. In Table 6, we show the zero-shot ImageNet, CIFAR-100 and Oxford-IIIT Pets classification accuracy for `HET-XL` versus a baseline `DET` model. We see a small but significant gain on the primary ImageNet accuracy metric and no significant difference in the CIFAR-100 and Oxford-IIIT Pets accuracies. To contextualize the magnitude of the gains, we note that when the same `DET` model is trained for 4 billion rather than 18 billion examples the zero-shot accuracy is 84.98/83.14/97.57 for ImageNet, CIFAR-100 and Oxford-IIIT Pet respectively. The gains we see from `HET-XL` on ImageNet are similar in magnitude to increasing the number of training examples shown to `DET` by a factor of 4.5.

To clarify the computation of the p-values: we compute 3 separate p-values using 3 separate two-sample unequal variance t-tests. We compute a p-value comparing the `DET` vs. `HET-XL` Imagenet zero-shot accuracy, a separate p-value for the CIFAR-100 results and similarly for Oxford-IIIT Pet. Only the ImageNet results are significant at the 0.01 level. Note however that the other two computed p-values are well above this level of significance. For clarity we report here the CIFAR-100 p-value: 0.839 and Oxford-IIIT Pet p-value: 0.596.

In Table 7 we compare the effect of different placements of the normalization operations in `HET-XL` for contrastive learning. Pre-normalization=True, means that the normalization operation is applied to the deterministic representation outputted by the image or text tower. Post-normalization=True means that the normalization operation is applied after the sampled noise vector has been added. We train with a ViT-L/32 text tower for 2 billion examples. We observe that it is crucial to add the noise term after normalization, however re-applying normalization after the noise has been added does not have a consistent effect. In the full scale experiments, Table 6 we apply normalization before adding the noise but not after i.e. pre-normalization=True and post-normalization=False.

## A.3 BACKGROUND ABOUT THE "HASHING TRICK"

It can be cumbersome to deal with the one-hot, or bag-of-word, encoding of features that have many categorical attributes. A case in point is large-scale advertising problems (Chapelle et al., 2014) with categorical features describing the large sets of publishers and advertisers. In such settings, we need to carefully bookkeep the different categories based on some associative data structures and it is difficult to control the space taken by the resulting feature representations (i.e., $\sum_{f \in \mathcal{F}} C_f$ where $C_f$ stands for the number of categories for the feature $f$ and $\mathcal{F}$ is the set of all the features).

The "hashing trick" (Weinberger et al., 2009; Chen et al., 2015; Eban et al., 2020) was developed to provide a simple-to-implement, robust[1], fast and space-efficient technique to deal with the above setting. The "hashing trick" considers

- A user-defined parameter $H$ explicitly controlling the space of the representation, with possibly $H \ll \sum_{f \in \mathcal{F}} C_f$. Here, $H$ will correspond to the number of hash buckets.
- A hashing function `hash_fn` mapping a pair of feature-value $(f, v_f)$ to $\{1, \ldots, H\}$.

The $H$-dimensional feature representation $\boldsymbol{\psi} \in \mathbb{R}^H$ of the raw data $\{(f, v_f)\}_{f \in \mathcal{F}}$ is obtained by the following simple procedure (see Section 4.3 in Chapelle et al. (2014)):

- Initialize $\boldsymbol{\psi} = \mathbf{0}$.
- For $f \in \mathcal{F}$, update $\boldsymbol{\psi}_j = \boldsymbol{\psi}_j + 1$ with $j = \text{hash\_fn}((f, v_f))$.[2]

Depending on the choice of $H$, the hashing trick operates as a dimensionality-reduction technique. It leads to *collisions* wherein two different feature-value pairs $(f, v_f)$ and $(f', v_{f'})$ end up with the same entry in the feature representation. In the example of large-scale advertising problems, the collisions generally act as a regularizer (Chapelle et al., 2014). This technique works well in practice and has been widely adopted.

**"Hashing trick" for `HET`.** Equipped with the previous insights, we can use the hashing trick to have a fast and space-efficient mapping from the space of labels $\mathbb{R}^K$ to $\mathbb{R}^H$. We can then define the covariance matrix into the resulting lower-dimensional space $\mathbb{R}^H$, which is precisely what `HET-H` in Section 6.1 does. Interestingly, in the context of `HET-H`, collisions have a different meaning: If two classes $c$ and $c'$ are mapped to the same entry in $\mathbb{R}^H$, then they will share their covariance structure. Our experiments with $H \ll K$ showed that those collisions did not seem to be harmful in our settings.

# B    LEARNING $\tau$

We ablate the effect of learning $\tau$ in the `HET-XL` method and also evaluate whether adding a learned temperature parameter to `DET` and `HET` makes up the gap with `HET-XL`, see Table 8. We observe that on JFT-300M and JFT-4B all methods benefit from having a learned temperature parameter, however the gains from `HET-XL` can only be partially explained by learning $\tau$. On ImageNet-21k, where we train for 90 epochs, only `HET-XL` sees a benefit from learning $\tau$, as opposed to applying a grid search based on the validation-set performance.

In the main paper, we argue that $\tau$ can be learned in the large-scale settings considered in the experiments. By "large-scale" settings, we mean that we are in a regime whereby (i) the data scale is large relative to the model capacity and (ii) the training schedule, as typical at the scale, entails a few epochs. Overfitting is therefore unlikely to happen.

Above, we observed on ImageNet-21k, with models trained for 90 epochs, that learning $\tau$ only has a positive effect for `HET-XL` and not the other methods. We hypothesise that this may indicate that this setup is already at the limit of the large-scale setting we described.

We evaluate our hypothesis that learning $\tau$ helps more when we can only train for a small number of epochs by comparing training on ImageNet-21k for 7 and 90 epochs, Table 9. We observe, as expected, that as we train for longer the gains from learning $\tau$ reduce.

---

[1]For example, the hashing trick can easily deal with new features and it can handle any feature type that is hashable.
[2]Sometimes, signed variants of the hashing trick are used (Weinberger et al., 2009).

Table 8: Ablation on the effect of learning $\tau$, all results for ResNet152x1 models.

| | JFT-300M | | ImageNet-21k | | JFT-4B | |
|---|---|---|---|---|---|---|
| Method | ↓NLL | ↑Prec@1 | ↓NLL | ↑Prec@1 | ↓NLL | ↑Prec@1 |
| DET | 8.57 | 0.429 | 5.79 | 0.461 | 5.44 | 0.508 |
| DET w/ learn $\tau$ | 8.44 | 0.449 | 5.82 | 0.458 | 5.39 | 0.514 |
| HET | 8.45 | 0.444 | 5.72 | 0.475 | - | - |
| HET w/ learn $\tau$ | 8.28 | 0.461 | 5.72 | 0.475 | - | - |
| HET-XL w/o learn $\tau$ | 8.15 | 0.460 | 5.72 | 0.475 | 5.37 | 0.513 |
| HET-XL | 8.11 | 0.467 | 5.71 | 0.486 | 5.34 | 0.517 |

Table 9: Evaluating the effect of learning $\tau$ as we reduce the number of training epochs on ImageNet-21k with a ViT-L/32 HET-XL model.

| Method | ↓NLL | ↑Prec@1 |
|---|---|---|
| 90 epochs grid search $\tau$ | 5.80 | 0.471 |
| 90 epochs w/ learn $\tau$ | 5.78 | 0.474 |
| 7 epochs grid search $\tau$ | 7.988 | 0.232 |
| 7 epochs w/ learn $\tau$ | 7.507 | 0.274 |

## C  MODEL SCALING ABLATION

We evaluate whether gains from the HET-XL method persist when the base model scale is doubled. We see in Table 5 that when the base model scale is increased from a ResNet152x1 to a ResNet152x2 the relative gains from the HET-XL method on JFT-300M are similar to those observed earlier with a ResNet152x1 Table 3.

## D  DATA SCALING ABLATION

In Table 4 (main paper) and Table 10, we see that increasing the number of training epochs has little effect on the relative gains from the HET-XL method, while increasing the performance of all methods. We scale training from 7 to 14 and 28 epochs on JFT-300M with the ViT-L/32 architecture, Table 4. Similarly we scale the training epochs from 1 to 2 and 3 epochs on the JFT-4B dataset with a ResNet152x1 architecture, Table 10.

Table 10: Effect of data scaling: ResNet152x1 JFT-4B comparison of methods. We report test set negative log-likelihood and precision@1.

| | 1 epoch | | 2 epochs | | 3 epochs | |
|---|---|---|---|---|---|---|
| Method | ↓NLL | ↑Prec@1 | ↓NLL | ↑Prec@1 | ↓NLL | ↑Prec@1 |
| DET | 5.44 | 0.508 | 5.38 | 0.515 | 5.35 | 0.518 |
| HET-XL | 5.34 | 0.517 | 5.28 | 0.523 | 5.25 | 0.526 |

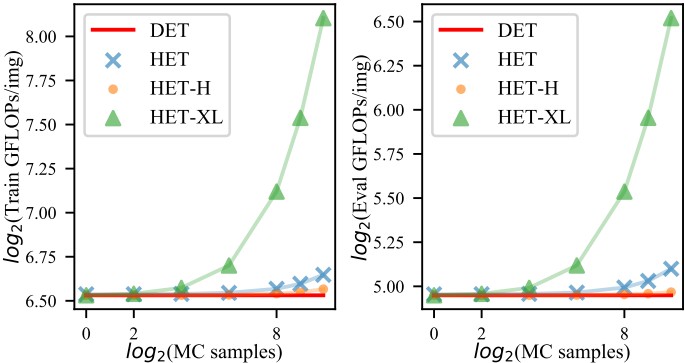

Figure 3: JFT-4B ViT-L/32 latency in terms of FLOPs.

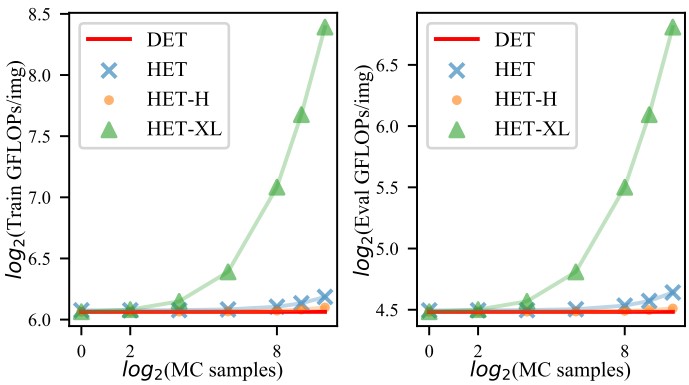

Figure 4: ImageNet-21k ResNet152x1 latency in terms of FLOPs.

## E  FLOPs METRIC

In the main paper, we report training and evaluation ms/img as the latency metric, Fig. 2. FLOPs is another popular metric evaluating the computational cost of neural network models. In Fig. 3 and Fig. 4, we report the training and evaluation FLOPs for a ViT-L/32 on JFT-4B and a ResNet152x1 on ImageNet-21k respectively. We see a similar trend as for ms/img. At high numbers of MC samples, HET-XL has a significantly higher FLOPs versus the key baseline, HET. However the trend is reversed at lower numbers of MC samples. In Table 14 of Appendix H, we will see that the predictive performance of HET-XL is not highly sensitive to the number of MC samples.

## F  FURTHER LATENCY AND MEMORY RESULTS

In the main paper, we only report memory usage and latency metrics in the ResNet152x1 ImageNet-21k setting, see Fig. 2. In Fig. 5 and Fig. 6, we additionally report memory usage and latency in the ViT-L/32 JFT-4B setting. We see similar trends as for the ImageNet-21k setting, with HET-XL memory usage less that HET and the ordering of the methods dependent on the number of MC samples.

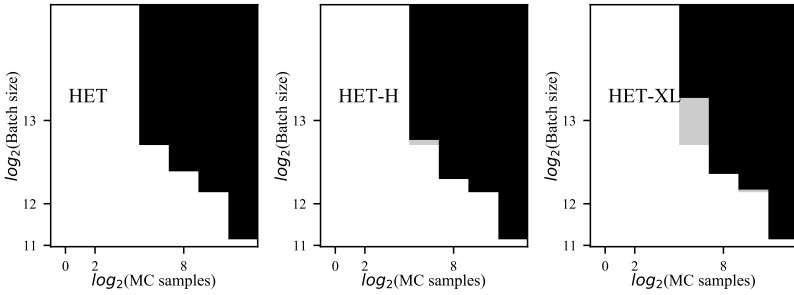

Figure 5: JFT-4B ViT memory usage. A black rectangle means that for the given number of MC samples and batch size the method goes out of memory. Plotted in grey are the cells where `HET-H` and `HET-XL` fit in memory but `HET` does not.

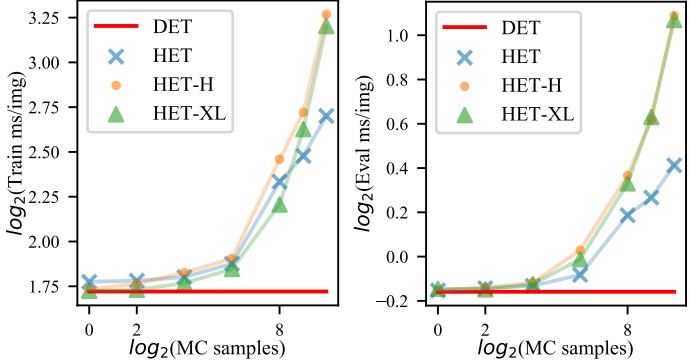

Figure 6: JFT-4B ViT-L/32 latency.

### F.1 AN ANALYSIS OF THE COMPUTATIONAL DEPENDENCY ON THE NUMBER OF MC SAMPLES

As previously observed, whether HET-XL has a better latency than HET depends on the number of MC samples used. We next study this observation by deriving the time complexity for both HET-XL and HET, showing that the comparison depends on a trade-off with respect to $K, D, R$ (as defined in Section 2.2) and $S$, which stands for the number of MC samples.

Let us define the dimension $Q$ equal to $K$ in the case of HET and equal to $D$, the dimension of the pre-logits, otherwise. With the notation from Section 2.2, we recall that $\phi(\boldsymbol{x}) \in \mathbb{R}^D$ and $\boldsymbol{J} \in \mathbb{R}^{R \times Q}$ where $R$ stands for the number of factors of the low-rank parametrization. For both HET-XL and HET and a number $S$ of MC samples, the noise matrices $\boldsymbol{\epsilon}_S(\boldsymbol{x}) \in \mathbb{R}^{S \times K}$ and $\boldsymbol{\epsilon}'_S(\boldsymbol{x}) \in \mathbb{R}^{S \times D}$ are generated according to (we omit all the bias terms for simplicity)

$$(\mathbf{1}_S \, \boldsymbol{v}(\boldsymbol{x})^\top) \circ (\boldsymbol{Z}\boldsymbol{J}) + (\mathbf{1}_S \, \phi(\boldsymbol{x})^\top \boldsymbol{K}_1) \circ (\boldsymbol{z} \, \mathbf{1}_Q^\top) \quad \text{with} \quad \boldsymbol{v}(\boldsymbol{x}) = \boldsymbol{K}_2^\top \phi(\boldsymbol{x}) \in \mathbb{R}^Q$$

and

- $\boldsymbol{z} \in \mathbb{R}^S \sim \mathcal{N}(\mathbf{0}, \boldsymbol{I})$,
- $\boldsymbol{Z} \in \mathbb{R}^{S \times R} \sim \mathcal{N}(\mathbf{0}, \boldsymbol{I})$,
- $\boldsymbol{K}_1 \in \mathbb{R}^{D \times Q}$,
- $\boldsymbol{K}_2 \in \mathbb{R}^{D \times Q}$.

Once the noise matrices $\boldsymbol{\epsilon}_S(\boldsymbol{x})$ and $\boldsymbol{\epsilon}'_S(\boldsymbol{x})$ have been generated, the logits for the $S$ MC samples are obtained by computing

$$\mathbf{1}_S \phi(\boldsymbol{x})^\top \boldsymbol{W} + \boldsymbol{\epsilon}_S(\boldsymbol{x}) \in \mathbb{R}^{S \times K} \quad \text{and} \quad (\mathbf{1}_S \phi(\boldsymbol{x})^\top + \boldsymbol{\epsilon}'_S(\boldsymbol{x}))\boldsymbol{W} \in \mathbb{R}^{S \times K} \tag{10}$$

for respectively HET and HET-XL.

In Table 11, we derive step by step the resulting time complexity for both HET and HET-XL, and show that we recover the experimental observation that HET-XL is more efficient than HET for small numbers of MC samples.

| Operations | Complexity HET | Complexity HET-XL |
|:---:|:---:|:---:|
| $\boldsymbol{v}(\boldsymbol{x})$ | $\mathcal{O}(DK)$ | $\mathcal{O}(D^2)$ |
| $(\mathbf{1}_S \, \boldsymbol{v}(\boldsymbol{x})^\top) \circ (\boldsymbol{Z}\boldsymbol{J})$ | $\mathcal{O}(KS + KRS)$ | $\mathcal{O}(DS + DRS)$ |
| $(\mathbf{1}_S \, \phi(\boldsymbol{x})^\top \boldsymbol{K}_1) \circ (\boldsymbol{z} \, \mathbf{1}_Q^\top)$ | $\mathcal{O}(DK + KS)$ | $\mathcal{O}(D^2 + DS)$ |
| Logits Eq. (10) | $\mathcal{O}(DK + KS)$ | $\mathcal{O}(DS + DKS)$ |
| Total | $\mathcal{O}(DK + KRS)$ | $\mathcal{O}(DRS + D^2 + DKS)$ |

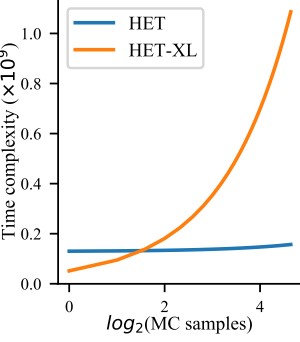

Table 11: Time complexity analysis of HET and HET-XL. For the total complexity in the last row, we only keep dominating terms, without constants if repeated multiple times (assuming $K \geq 1$, $S \geq 1$ and $R \geq 1$). We additionally plot on the right figure the non-simplified time complexities of HET and HET-XL, instantiated with the values of the ResNet152x1 setting on ImageNet-21k. We do observe the phenomenon wherein HET-XL is more efficient for fewer MC samples. The crossing point is not exactly the same as that in the real experiments because of the big-$\mathcal{O}$ analysis and some difficult-to-model linear-algebra optimization due to XLA (Sankaran et al., 2022).

We note that this time complexity analysis agrees with the experimentally observed training times computed in our profiling. In particular Table 12 shows that on ImageNet-21k with ResNet152x1 the training time for HET-XL is faster than HET up to 16 MC samples, with HET being faster from 64 MC samples on. Note that there are several points at which HET-XL dominates HET e.g. HET-XL has better Prec@1 and NLL at 1 MC sample than HET at 1000 MC samples despite also being substantially faster to train. Training times are computed from by multiplying the profiler train ms/img by the number of examples seen during training, as commonly done to benchmark large-scale vision models (Riquelme et al., 2021).

Table 12: TPU training time vs. performance for varied MC samples for `HET` and `HET-XL` on ResNet152x1 ImageNet-21k.

| MC Samples | 1 | 4 | 16 | 64 | 256 | 512 | 1000 |
|---|---|---|---|---|---|---|---|
| `HET` Prec@1 | 0.464 | 0.433 | 0.462 | 0.466 | 0.469 | 0.472 | 0.475 |
| `HET` NLL | 5.80 | 6.00 | 5.77 | 5.75 | 5.73 | 5.73 | 5.72 |
| `HET` TPU Training Hours | 1217 | 1221 | 1230 | 1263 | 1547 | 1666 | 1786 |
| `HET-XL` Prec@1 | 0.476 | 0.478 | 0.476 | 0.480 | 0.485 | 0.484 | 0.486 |
| `HET-XL` NLL | 5.69 | 5.68 | 6.12 | 5.99 | 5.79 | 5.70 | 5.71 |
| `HET-XL` TPU Training Hours | 1171 | 1181 | 1205 | 1316 | 1745 | 2244 | 3278 |

## G  TUNING OF $\tau$

In this section, we discuss further details and results of Section 4.

### G.1  DETAILS ABOUT THE BENCHMARK

We consider the tuning of `HET-XL` for a ResNet50x2 model over ImageNet-21k with a training duration of 7 epochs. The hyperparameters of the model are those from Table 3 in Dosovitskiy et al. (2020).

To define our 3-fold split of the dataset, we take the standard validation set as the test set (containing 102,400 points), and extract from the training set a validation set (also with 102,400 points). All the methods in the benchmark are tuned to minimize the validation negative log likelihood.[3]

The grid search uses the following list of temperatures $\tau \in \{0.05, 0.1, 0.2, 0.4, 0.8, 1.5, 3.0, 5.0\}$ which cover typical well-performing values mentioned in previous work (Collier et al., 2021).

The Bayesian optimization approach follows the default configuration from Song et al. (2022). We run a total of 30 training jobs, divided into 6 sequential steps of 5 parallel training jobs.

As far as the `HET-XL`-related hyperparameters are concerned, we set the range of $\tau$ as $[0.05, 5.0]$, consider $R = 50$ factors and use 1000 MC samples, as done in the rest of the paper.

### G.2  THE SPECIFIC STRUCTURE OF THE HYPERPARAMETER $\tau$

The temperature $\tau$ has the property to explicitly appear both at training and validation time, unlike optimisation-related or regularisation-related hyperparameters that surface only at training time. As we next see, this observation has implications about the way we can compute a gradient estimator to tune $\tau$.

To ease the exposition, let us momentarily simplify our setup. We consider a temperature-scaled model $1/\tau \cdot f(\boldsymbol{x}; \Theta) \in \mathbb{R}$ with a one-dimensional output, e.g., in a regression task. The parameters of the model are summarised in $\Theta \in \mathbb{R}^P$. Moreover, let $\mathcal{L}_{\text{train}}(\cdot, \cdot)$ and $\mathcal{L}_{\text{val}}(\cdot, \cdot)$ be two loss functions, defined respectively for training data $(\boldsymbol{x}, y) \sim p_{\text{train}}$ and validation data $(\boldsymbol{x}, y) \sim p_{\text{val}}$.

We typically seek a temperature parameter that minimises some validation objective in a bi-level optimisation formulation (Franceschi et al., 2018)

$$\min_{\tau > 0} F_{\text{val}}(\tau) \quad \text{with} \quad F_{\text{val}}(\tau) = \mathbb{E}_{(\boldsymbol{x}, y) \sim p_{\text{val}}} \left[ \mathcal{L}_{\text{val}} \left( \frac{1}{\tau} \cdot f(\boldsymbol{x}; \Theta^*(\tau)), y \right) \right] \tag{11}$$

---

[3]There is one exception: We also check in Table 13 the effect of applying grid search and Bayesian optimization directly to optimize for the test metric. Those variants are referred to as "2-fold".

for some parameters $\Theta^*(\tau)$ defined as a solution of the training problem

$$\Theta^*(\tau) \in \underset{\Theta \in \mathbb{R}^P}{\arg \min} \; F_{\text{train}}(\tau, \Theta) \quad \text{with} \quad F_{\text{train}}(\tau, \Theta) = \mathbb{E}_{(\boldsymbol{x}, y) \sim p_{\text{train}}}\left[\mathcal{L}_{\text{train}}\left(\frac{1}{\tau} \cdot f(\boldsymbol{x}; \Theta), y\right)\right]. \quad (12)$$

By applying the chain rule, we can compute the gradient of the objective in Eq. (11) for a given pair $(\boldsymbol{x}, y)$ of the expectation, leading to:

$$\frac{\partial \mathcal{L}_{\text{val}}(y', y)}{\partial y'}\bigg|_{y' = \frac{1}{\tau} f(\boldsymbol{x}; \Theta^*(\tau))} \cdot \left\{ -\frac{1}{\tau^2} f(\boldsymbol{x}; \Theta^*(\tau)) + \frac{1}{\tau} \langle \nabla_\Theta f(\boldsymbol{x}; \Theta)|_{\Theta = \Theta^*(\tau)}, \nabla_\tau \Theta^*(\tau) \rangle \right\}. \quad (13)$$

In the expression above, the most complex term to handle is $\nabla_\tau \Theta^*(\tau)$ since it requires to differentiate through the solution of another optmization problem, notably involving higher-order derivatives (Maclaurin et al., 2015; Luketina et al., 2016; Pedregosa, 2016; Franceschi et al., 2018; Lorraine et al., 2020).

Several approaches have been proposed to approximate this term. For example, Luketina et al. (2016) only consider the hyperparameter influence through the current gradient step solving Eq. (12). This formally translates into $\nabla_\tau \Theta^*(\tau) \approx -s_t \cdot \nabla_\tau[\nabla_\Theta F_{\text{train}}(\tau, \Theta_t)]$ with $s_t$ and $\Theta_t$ being respectively the learning rate and iterate produced at step $t$ by an iterative algorithm solving Eq. (12), e.g., SGD. In our comparison, we will take Luketina et al. (2016) as a representative approach for efficient gradient-based hyperparameter tuning.

**A simple gradient estimator.** A remarkable property of the temperature is that $F_{\text{val}}$ in Eq. (11) depends on $\tau$ not only through $\Theta^*(\tau)$—as it would be for example the case for weight decay—but also directly via $1/\tau$. While in absence of this property, the gradient of $F_{\text{val}}$ would solely rely on $\nabla_\tau \Theta^*(\tau)$, in the case of $\tau$, we can assume that $\nabla_\tau \Theta^*(\tau) = 0$ (i.e., the optimal $\Theta^*(\tau)$ mildly varies with $\tau$) and focus on $-\frac{1}{\tau^2} f(\boldsymbol{x}; \Theta^*(\tau))$ for the gradient. Interestingly, further approximating $\Theta^*(\tau) \approx \Theta_t$, the resulting estimator is extremely simple and cheap to compute, e.g., no higher-order derivatives to compute (in Table 13, we refer to this approach as "Simple gradient"). Moreover, we can notice that, provided that $\mathcal{L}_{\text{train}} = \mathcal{L}_{\text{val}}$ (in our comparison, both are set to the negative log likelihood) and the training and validation sets are identical, $-\frac{1}{\tau^2} f(\boldsymbol{x}; \Theta_t)$ falls back to the gradient we would take by treating $\tau$ as a standard training parameter by directly solving $\min_{\tau > 0, \Theta \in \mathbb{R}^P} F_{\text{train}}(\tau, \Theta)$.

## G.3 ADDITIONAL OBSERVATIONS

In addition to the results reported and discussed in Section 4, we make the following observations from Table 13:

- In the training regime we are focusing on, it appears that applying grid search and Bayesian optimization to directly optimize the test likelihood ("2 fold") does not lead to significantly better performances.

- Retraining the best models obtained by grid search and Bayesian optimization ("+retrain") typically comes with a drop in performance, which can be explained by the mismatch of the selected temperature when the training set is enlarged.

- The coarser approximation made by "Simple gradient" compared with Luketina et al. (2016) leads to a worse likelihood.

- The approach of Luketina et al. (2016) suffers more than "Simple gradient" when we approximate the true validation set by a fraction of the training batch (here, 12.5% and 25%). It may be explained by the fact that "Simple gradient" reduces to "Simple training" in the case where the validation and training sets coincide. Note that, when 12.5% and 25% of the training batch are held-out as validation data, the model effectively sees fewer training points. We account for this effect by running for respectively 12.5% and 25% more epochs.

Table 13: Comparisons (means $\pm$ standard errors) of methods to tune $\tau$ for a ResNet50x2 on ImageNet-21k.

| Method | ↓NLL | ↑Prec@1 | Single training? | No validation set needed? |
|---|---|---|---|---|
| Grid search | 6.08 $_{\pm\,0.04}$ | 0.433 $_{\pm\,0.002}$ | ✗ | ✗ |
| Grid search (2 folds) | 6.11 $_{\pm\,0.05}$ | 0.434 $_{\pm\,0.001}$ | ✗ | ✗ |
| Grid search (+ retrain) | 6.11 $_{\pm\,0.05}$ | 0.434 $_{\pm\,0.001}$ | ✗ | ✗ |
| Bayesian optimization | 6.03 $_{\pm\,0.06}$ | 0.428 $_{\pm\,0.000}$ | ✗ | ✗ |
| Bayesian optimization (2 folds) | 6.05 $_{\pm\,0.07}$ | 0.435 $_{\pm\,0.001}$ | ✗ | ✗ |
| Bayesian optimization (+ retrain) | 6.09 $_{\pm\,0.05}$ | 0.429 $_{\pm\,0.001}$ | ✗ | ✗ |
| Luketina et al. (2016) | 6.08 $_{\pm\,0.04}$ | 0.425 $_{\pm\,0.000}$ | ✓ | ✗ |
| Luketina et al. (2016) (12.5%) | 6.19 $_{\pm\,0.07}$ | 0.423 $_{\pm\,0.001}$ | ✓ | ✓ |
| Luketina et al. (2016) (25.0%) | 6.20 $_{\pm\,0.07}$ | 0.420 $_{\pm\,0.001}$ | ✓ | ✓ |
| Simple gradient (val) | 6.22 $_{\pm\,0.05}$ | 0.432 $_{\pm\,0.001}$ | ✓ | ✗ |
| Simple gradient (12.5%) | 6.18 $_{\pm\,0.04}$ | 0.432 $_{\pm\,0.001}$ | ✓ | ✓ |
| Simple gradient (25.0%) | 6.16 $_{\pm\,0.05}$ | 0.429 $_{\pm\,0.001}$ | ✓ | ✓ |
| Simple training (as in HET-XL) | 5.99 $_{\pm\,0.04}$ | 0.435 $_{\pm\,0.001}$ | ✓ | ✓ |

Table 14: Sensitivity analysis of HET-XL to the number of MC samples on ImageNet-21k with ResNet152x1 architecture.

| MC Samples | 1 | 4 | 16 | 64 | 256 | 512 | 1000 |
|---|---|---|---|---|---|---|---|
| Prec@1 | 0.476 | 0.478 | 0.476 | 0.480 | 0.485 | 0.484 | 0.486 |
| NLL | 5.69 | 5.68 | 6.12 | 5.99 | 5.79 | 5.70 | 5.71 |

## H   HET-XL SENSITIVITY TO THE NUMBER OF MC SAMPLES

We have seen above that the latency of the HET-XL method is sensitive to the number of MC samples used at training and eval time. In Table 14, we conduct a sensitivity analysis of the method's predictive performance to the number of MC samples. We see that while, as expected, more MC samples results in broadly better predictive performance, the number of training and evaluation samples can even be reduced to 1 for HET-XL and still outperform HET with 1,000 MC samples.

## I   EFFECT OF SHARING $W$ PROJECTION WITH NOISE SAMPLES

Three possible explanations for HET-XL's outperformance of HET are (i) learning $\tau$, (ii) knowledge sharing due to the use of $W$ to project both the pre-logits and the noise to logit space and (iii) the regularizing effect of vastly reducing the parameter count. We have already seen, Table 8 Appendix B, that learning the temperature only accounts for part of the gains. Below, Table 15, we run an ablation to test (ii) whether the $W$ sharing has a significant effect on HET-XL's performance. When we do not share the $W$, we instantiate a separate independent randomly initialized matrix $W'$ which is used solely to transform the noise samples to logit space; formally,

$$p(y|\boldsymbol{x}) = \mathbb{E}_{\boldsymbol{\epsilon}'}\left[\sigma\left(\boldsymbol{W}^\top\phi(\boldsymbol{x};\theta) + (\boldsymbol{W}')^\top\boldsymbol{\epsilon}'(\boldsymbol{x})\right)\right] \quad \text{with} \quad \boldsymbol{\epsilon}'(\boldsymbol{x}) \in \mathbb{R}^M \quad \text{and} \quad \boldsymbol{W}' \in \mathbb{R}^{M\times K}, \quad (14)$$

and $M \neq D$.

Table 15: Effect of sharing $\boldsymbol{W}$ projection between the pre-logits $\phi(\boldsymbol{x};\theta)$ and noise samples $\boldsymbol{\epsilon}(\boldsymbol{x})$. Evaluated on ImageNet-21k with ResNet152x1 architecture.

| Method | $\downarrow$**NLL** | $\uparrow$**Prec@1** |
|---|---|---|
| HET-XL w/ $\boldsymbol{W}$ sharing | 5.71 | 0.486 |
| HET-XL w/o $\boldsymbol{W}$ sharing | 5.62 | 0.487 |

Table 16: Sensitivity to $M$ of ResNet152x1 ImageNet-21k performance when not sharing $\boldsymbol{W}$.

| $M$ | **64** | **128** | **256** | **512** | **1024** | **2048** |
|---|---|---|---|---|---|---|
| Prec@1 | 0.465 | 0.468 | 0.472 | 0.477 | 0.484 | 0.487 |
| NLL | 5.74 | 5.68 | 5.66 | 5.66 | 5.74 | 5.62 |

In the setting with a ResNet152x1 on ImageNet-21k, we can see that sharing $\boldsymbol{W}$ has a small, if anything slightly negative effect on performance. Hence, we conclude that the primary drivers of HET-XL's improved performance are learning $\tau$ and the regularization due to HET-XL's reduced parameter count. For the HET-XL method presented in the other experiments throughout this paper, we opt to share $\boldsymbol{W}$ due to the minimal performance difference compared to not sharing $\boldsymbol{W}$ as well as the added simplicity and efficiency of the method as a result.

HET-XL ties the dimensionality of $\boldsymbol{\epsilon}'(\boldsymbol{x})$ to the dimensionality of the last layer of the base network, $D$. Following Eq. (14), this is not strictly necessary and we could allow the dimension of $\boldsymbol{\epsilon}'(\boldsymbol{x})$ to be independent of $D$. In Table 16 we vary precisely $M$. Note that $M$ is different from $R$, the rank of $\boldsymbol{\Sigma}'(\boldsymbol{x};\theta_{\text{cov}})$. We observe that indeed as we increase $M$ to $D$ we see improved Prec@1 and NLL. Note however that we can decrease $M$ by a factor of 8 from $D$ down to $M = 256$, and still have performance better than HET.

## I.1 ANOTHER MOTIVATION FOR THE SHARING OF $W$

We provide another observation that further motivates why sharing the same matrix $\boldsymbol{W}$ for both $\phi(\boldsymbol{x};\theta)$ and $\boldsymbol{\epsilon}'(\boldsymbol{x})$, as done by HET-XL in Eq. (5), is a natural design decision

Let us write the rank-$R$ covariance matrices for both HET and HET-XL

$$\boldsymbol{\Sigma}_{\text{HET}}(\boldsymbol{x}) = \boldsymbol{Q}(\boldsymbol{x})^\top \boldsymbol{Q}(\boldsymbol{x}) \quad \text{with} \quad \boldsymbol{Q}(\boldsymbol{x}) \in \mathbb{R}^{R\times K}$$

and

$$\boldsymbol{\Sigma}_{\text{HET-XL}}(\boldsymbol{x}) = \boldsymbol{W}^\top \boldsymbol{P}(\boldsymbol{x})^\top \boldsymbol{P}(\boldsymbol{x})\boldsymbol{W} \quad \text{with} \quad \boldsymbol{P}(\boldsymbol{x}) \in \mathbb{R}^{R\times D}, \boldsymbol{W} \in \mathbb{R}^{D\times K}.$$

A sufficient condition for HET-XL to be able to express a covariance similar to that of HET is to have

$$\boldsymbol{Q}(\boldsymbol{x})^\top \approx \boldsymbol{W}^\top \boldsymbol{P}(\boldsymbol{x})^\top. \tag{15}$$

In other words, if the span of the rows of $\boldsymbol{W}$—a subspace of $\mathbb{R}^K$ also referred to as row space of $\boldsymbol{W}$ or $\texttt{rowsp}(\boldsymbol{W})$ for short—is well aligned with $\texttt{rowsp}(\boldsymbol{Q}(\boldsymbol{x}))$, we can find a matrix $\boldsymbol{P}(\boldsymbol{x})$ to satisfy the above relationship.

We test the hypothesis that the $\boldsymbol{W}$ learned by HET naturally aligns with $\texttt{rowsp}(\boldsymbol{Q}(\boldsymbol{x}))$, which would justify our design choice for HET-XL. To this end, we consider the HET ResNet152x1 model trained on ImageNet-21k. To quantitatively measure the alignment of $\texttt{rowsp}(\boldsymbol{W})$ and $\texttt{rowsp}(\boldsymbol{Q}(\boldsymbol{x}))$, we use the *smallest principal angle* (SPA) between those two subspaces (Björck & Golub, 1973). We will manipulate the SPA via its cosine, i.e., $\cos(a_{\text{SPA}}(\boldsymbol{x}))$. Computing the SPA requires to compute the SVD of $\boldsymbol{B}_{\boldsymbol{W}}^\top \boldsymbol{B}_{\boldsymbol{Q}(\boldsymbol{x})}$ with $\boldsymbol{B}_*$ denoting orthonormal basis for $\texttt{rowsp}(\boldsymbol{W})$ and $\texttt{rowsp}(\boldsymbol{Q}(\boldsymbol{x}))$ (Björck & Golub, 1973).

Table 17: Measurement of the alignment between $\texttt{rowsp}(\boldsymbol{W})$ and $\texttt{rowsp}(\boldsymbol{Q}(\boldsymbol{x}))$ for HET. Evaluated on ImageNet-21k with a ResNet152x1 architecture. The table reports the average and the standard deviation of the (cosine of the) smallest principal angle (SPA) over the validation set. A higher cosine implies a *smaller* angle and a better alignment of $\texttt{rowsp}(\boldsymbol{W})$ and $\texttt{rowsp}(\boldsymbol{Q}(\boldsymbol{x}))$. Here, we have $\arccos(0.829) \approx 0.593 < \arccos(0.350) \approx 1.213$.

| Setting | $\mathbb{E}_{\boldsymbol{x}}[\cos(a_{\mathrm{SPA}}(\boldsymbol{x}))]$ |
|---|---|
| With $\boldsymbol{W}$ learned by HET | $0.829 \pm 0.042$ |
| With random $\boldsymbol{W}$ | $0.350 \pm 0.002$ |

Table 18: Ablation on the effect of the dynamics of $\tau$ throughout training on predictive performance. We first train a HET-XL model and then train a new model from scratch fixing the temperature to be the final value of the learned temperature from the first training run. Evaluated on ImageNet-21k with a ResNet152x1 architecture.

| Method | $\downarrow$NLL | $\uparrow$Prec@1 |
|---|---|---|
| HET-XL | 5.71 | 0.486 |
| HET-XL w/ $\tau$ fixed to final value of HET-XL learned temperature | 5.74 | 0.486 |

We compute $\cos(a_{\mathrm{SPA}}(\boldsymbol{x}))$ for each image $\boldsymbol{x}$ of the validation set and report in Table 17 the resulting mean and standard deviation. To put in perspective the value obtained in the case of the learned $\boldsymbol{W}$ by HET, we also compute the same quantities in the case of randomly generated Gaussian $\boldsymbol{W} \sim \mathcal{N}(\boldsymbol{0}, \boldsymbol{I})$. We can see that in the case of HET, the SPA is significantly smaller (i.e., its cosine significantly larger) than in the case of a random $\boldsymbol{W}$, with $\arccos(0.829) \approx 0.593 < \arccos(0.350) \approx 1.213$. Compared with the random case, the learned $\boldsymbol{W}$ is thus such that $\texttt{rowsp}(\boldsymbol{W})$ and $\texttt{rowsp}(\boldsymbol{Q}(\boldsymbol{x}))$ are well aligned, as required by Eq. (15).

## J $\tau$ DYNAMICS

Fig. 7 shows the evolution of $\tau$ during training on ImageNet-21k with a ResNet152x1. It is interesting to note the non-monotonic dynamics of $\tau$. In Table 18, we take the temperature at the end of this training run, $\tau = 0.50940$ and re-run training from scratch with this fixed temperature value. The performance of the methods are similar, suggesting that the benefits from learning $\tau$ come primarily from finding a very fine-grained value of $\tau$ to train with and not that $\tau$ can take different values over time. Finding such specific values of $\tau$ would only be possible with an expensive fine-grained hyperparameter search, which is infeasible in the settings considered in this paper.

## K $\tau$ PARAMETERIZATION AND SENSITIVITY TO $\tau$ STARTING POINT AND RANGE

When learning the temperature parameter, we parameterize $\tau$ as

$$\tau(t) = (\tau_{\max} - \tau_{\min}) \cdot \texttt{sigmoid}(t) + \tau_{\min} \tag{16}$$

where $t$ is an unbounded trainable parameter to be learned and $\tau_{\min}$ and $\tau_{\max}$ are hyperparameters defining the valid range of minimumn and maximum possible temperatures. For all experiments in the main paper, we use $\tau_{\min} = 0.05$, $\tau_{\max} = 5.0$ and initialize $t$ such that the initial $\tau_0$ is set to the midpoint of $[0.05, 5.0]$, i.e., 2.525.

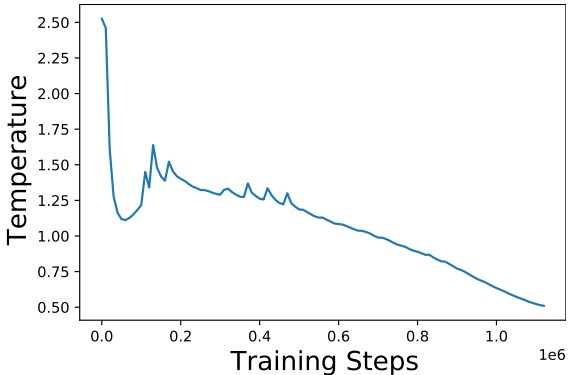

Figure 7: $\tau$ dynamics during training of ResNet152x1 on ImageNet-21k.

Table 19: Sensitivity to $\tau$ starting point. Evaluated on ImageNet-21k with ResNet152x1 architecture.

| $\tau_0$ | $\tau_{\min}$ | $\tau_{\max}$ | $\downarrow$NLL | $\uparrow$Prec@1 |
|------|------|-----|------|-------|
| 1.25 | 0.05 | 5.0 | 5.59 | 0.485 |
| 2.525 | 0.05 | 5.0 | 5.71 | 0.486 |
| 3.75 | 0.05 | 5.0 | 5.73 | 0.485 |
| 4.9 | 0.05 | 5.0 | 5.72 | 0.478 |

In Table 19 and Table 20 we evaluate the effect of the choice of $\tau_{\min}$ and $\tau_{\max}$ and the initialization point along this range. We see that the final performance in the ResNet152x1 ImageNet-21k setting is not sensitive to these choices and that, if anything, further small improvements to the results in the paper would be possible by tuning these hyperparameters. Since the removal of the need to tune the temperature hyperparameter in the `HET` method is the primary motivation for learning $\tau$, we are satisfied with the insensitivity in performance to the temperature range and starting point and simply use this good default of $\tau_{\min} = 0.05$, $\tau_{\max} = 5.0$ for all experiments.

## L    DIAGONAL COMPONENT PARAMETERIZATION

We recall that Collier et al. (2021) parameterize the covariance matrix as

$$\mathbf{\Sigma}(\boldsymbol{x}) = \boldsymbol{V}(\boldsymbol{x})^\top \boldsymbol{V}(\boldsymbol{x}) + \mathrm{diag}(\boldsymbol{d}(\boldsymbol{x})) \quad \text{with} \quad \boldsymbol{V}(\boldsymbol{x}) \in \mathbb{R}^{R \times K} \quad \text{and} \quad \boldsymbol{d}(\boldsymbol{x}) \in \mathbb{R}^K_+. \tag{17}$$

Table 20: Sensitivity to $\tau$ range. Evaluated on ImageNet-21k with ResNet152x1 architecture.

| $\tau_0$ | $\tau_{\min}$ | $\tau_{\max}$ | $\downarrow$NLL | $\uparrow$Prec@1 |
|------|-------|-----|------|-------|
| 2.525 | 0.05 | 5.0 | 5.71 | 0.486 |
| 2.525 | 0.001 | 5.0 | 5.50 | 0.488 |
| 2.525 | 0.01 | 5.0 | 5.73 | 0.485 |
| 2.525 | 0.05 | 4.0 | 5.95 | 0.485 |
| 2.525 | 0.05 | 3.0 | 5.61 | 0.485 |

Table 21: Comparing two parametrizations of the covariance matrix with equal parameter counts, namely $V(x)^\top V(x) + \text{diag}(d(x))$ versus $V(x)^\top V(x) + d(x)d(x)^\top$. Evaluated on ImageNet-21k with a ResNet152x1 architecture.

| Method | ↓NLL | ↑Prec@1 |
|---|---|---|
| HET-XL w/ $d(x)d(x)^\top$ | 5.71 | 0.486 |
| HET-XL w/ $\text{diag}(d(x))$ | 5.67 | 0.486 |
| HET-XL w/ $d(x)d(x)^\top$ - learn $\tau$ | 5.72 | 0.474 |
| HET-XL w/ $\text{diag}(d(x))$ - learn $\tau$ | 5.72 | 0.473 |

However, we observe that, for the same parameter count, we can increase the expressiveness of $\Sigma(x)$ while also reducing (marginally) the cost of MC sampling $\epsilon(x)$ by replacing the diagonal component with a rank-1 component

$$\Sigma(x) = V(x)^\top V(x) + d(x)d(x)^\top. \tag{18}$$

The marginally reduced cost of MC sampling is a consequence of the shape of underlying standard Normal samples $\mathcal{N}(0, I)$ required to sample from $\text{diag}(d(x))$ versus $d(x)d(x)^\top$. Let us denote by $S$ the number of MC samples. In the diagonal case, we must sample a tensor of standard Normal values of shape $[S, D]$, whereas similar to Monteiro et al. (2020), to induce the non-diagonal covariance terms only a tensor of shape $[1, D]$ is required. This represents a saving of $(S − 1) \times D$ standard Normal samples per example. In Table 21 we empirically evaluate these choices. We see that the choice of this rank-1 vs. diagonal component parameterization has little effect on the predictive performance. Hence, due to the marginal efficiency savings from the rank-1 parameterization, we choose to parametertize HET-XL accordingly. To ensure a fair comparison, we also modify the Collier et al. (2021) method parameterization to use this rank-1 approach. This is the HET method reported throughout the paper.

## M DETERMINISTIC VERSUS STOCHASTIC HETEROSCEDASTIC CONTRASTIVE LEARNING AT TEST TIME

In our contrastive learning setup, we use only 1 MC sample at training time. At test time we use the deterministic representations to perform zero-shot classification, despite having added heteroscedastic noise at training time. This is similar to the original HET paper, (Collier et al., 2021) which fine-tunes deterministic representations from the HET model downstream on the VTAB (Zhai et al., 2019) benchmark. Therefore, at test time the computational cost of HET-XL is equal to that of DET. However we could also perform MC sampling at test time. Here we ablate our choice by performing test time MC sampling with 22 MC samples (the maximum that fits in memory) and comparing the results to using the non-stochastic representations. We use a smaller scale setup with a ViT-L/32 text encoder and 8 billion training examples (all other hyperparameters as per the contrastive experiment in the main paper). We observe exactly the same zero-shot ImageNet accuracy of 84.35% regardless of whether we perform MC sampling or use the deterministic representation at test time.

## N EXPERIMENTAL DETAILS AND HYPERPARAMETERS

All hyperparameters are as follows unless otherwise stated.

### N.1 Image classification

All image classification experiments are trained on 64 TPU v3 cells with 128 cores. Heteroscedastic methods use 1,000 MC samples and the parameter efficient covariance parameterization with $R = 50$ is used. The temperature parameter for HET-XL is parameterized as per Appendix K.

**JFT-300M.** All methods and models are trained for 7 epochs with the Adam optimizer with $\beta_1 = 0.9$, $\beta_2 = 0.999$ and weight decay of 0.1 and otherwise default JAX hyperparameters. The learning rate undergoes a 10,000 linear warm-up phase starting at $10^{-5}$ and reaching $6 \times 10^{-4}$. A batch size of 4096 is used.

**ImageNet-21k.** All methods and models are trained for 90 epochs with the Adam optimizer with $\beta_1 = 0.9$, $\beta_2 = 0.999$ and weight decay of 0.03 for ResNet152 and 0.1 for ViT-L/32 and otherwise default JAX hyperparameters. The learning rate undergoes a 10,000 linear warm-up phase starting at $10^{-5}$ and reaching $10^{-3}$. A batch size of 1024 is used for ResNet152 and 4096 for ViT-L/32.

**JFT-4B.** All methods and models are trained for 1 epoch with the Adam optimizer with $\beta_1 = 0.9$, $\beta_2 = 0.999$ and weight decay of 0.1 and otherwise default JAX hyperparameters. The learning rate undergoes a 10,000 linear warm-up phase starting at $10^{-5}$ and reaching $6 \times 10^{-4}$. A batch size of 4096 is used.

We note that we reuse, without modification, **all** the hyperparameters of the HET method which have been tuned in prior work (Tran et al., 2022). The only hyperparameters introduced by the HET-XL method are the boundaries of the learned temperature parameter which we set to be min=0.05 and max=5.0 for all experiments. We find that performance is insensitive to these boundaries provided the boundaries contain the final temperature learned by HET-XL, see Appendix K. Therefore we conclude that HET-XL is relatively insensitive to hyperparameter choice, given that all existing hyperparameters can be reused and we remove the need for tuning the main sensitive hyperparameter from the HET method ($\tau$).

### N.2 Contrastive learning

All hyperparameters are taken from the original LiT paper Zhai et al. (2022) unless otherwise specified. We use a JFT pre-trained ViT-G image encoder and a ViT-g text encoder with a vocabulary size of 32,000. We train with a batch size of 16,384 for 18B examples on 256 v3 TPU cells (512 cores). A cosine learning rate schedule is used with a base learning rate of $10^{-3}$ and 10,000 warm-up steps. Weight decay of $10^{-4}$ is used. For HET-XL, the learned temperature is initialized to 3.0 and for DET it is initialized to 10.0. $R = 15$ for the non parameter-efficient covariance parameterization in HET-XL, which is trained with 1 MC sample but evaluated using the deterministic representations.

## O  Few-shot performance

We follow Tran et al. (2022) in evaluating the representations learned by HET-XL with a linear evaluation protocol on 9 few-shot learning datasets; birds, caltech, cars, cifar100, col hist, dtd, imagenet, pets, and uc merced; see Tran et al. (2022) for further details about the datasets. We evaluate 1-shot, 5-shot, 10-shot and 25-shot performance. We evaluate 9 upstream settings

- ViT-L/32 trained on JFT-4B for 1 epoch,
- ViT-L/32 trained on JFT-300M for 7, 14 and 28 epochs,
- ResNet152x1 trained on JFT-4B for 1, 2 and 3 epochs,
- ResNet152x1 trained on JFT-300M for 7 epochs and

Table 22: Average few-shot accuracy over 9 downstream datasets and 9 different upstream settings $\pm$ 1 std. error. An upstream setting is defined by an upstream dataset, a base architecture and a number of epochs. Please note that the standard errors are only equal after rounding and the reported standard errors are correct.

| Method | 1-shot ACC. | 5-shot ACC. | 10-shot ACC. | 25-shot ACC. |
|--------|-------------|-------------|--------------|--------------|
| DET    | 59.00 ($\pm$0.054) | 81.27 ($\pm$0.025) | 81.66 ($\pm$0.033) | 85.12 ($\pm$0.029) |
| HET-XL | **59.02** ($\pm$0.054) | **81.92** ($\pm$0.025) | **82.70** ($\pm$0.033) | **85.87** ($\pm$0.029) |

- ResNet152x2 trained on JFT-300M for 7 epochs.

Table 22 shows the performance for each number of shot(s), aggregated over the downstream datasets and the upstream settings. We can observe that in all settings HET-XL yields representations that transfer to downstream tasks better than DET.

## P    EQUALIZE DET PARAMS WITH HET-XL

Similar to (Collier et al., 2021) we implement an ablation in which the DET model's capacity is increased to the same number of parameters as HET-XL and the performance is compared. In particular, for JFT-300M trained for 7 epochs with a ViT-L/32 we increase the width of the last Dense layer in the network from 1,024 to 1,138. As a result the DET network has 327.5M parameters, the same number as the HET-XL network. The DET model with increased capacity achieves Prec@1: 0.470 and NLL: 7.82, a small improvement on the results in Table 3 which show the lower capacity DET model with Prec@1: 0.468 and NLL: 7.83, but still far behind the HET-XL model which achieves Prec@1: 0.498 and NLL: 7.65 at the same parameter count. This demonstrates that HET-XL's performance gains cannot simply be attributed to increased model capacity.

