# OpenReview forum: "Massively Scaling Heteroscedastic Classifiers"
_ICLR.cc/2023/Conference — ICLR 2023 poster_

### Official Review · Reviewer_FYog · 2022-10-23

**Confidence:** 3
**Correctness:** 3
**Technical Novelty And Significance:** 3
**Empirical Novelty And Significance:** 3
**Recommendation:** 6

**Clarity, Quality, Novelty And Reproducibility:**

- clarity: good
- quality:
    - while the idea of reusing the mean term is very straightforward and intuitive, it seems unintuitive to me that the idea can work very well
        - mathematically speaking or intuitive speaking, reusing the mean term should introduce identification issues in the estimation of the model, and the same effects (from feature to label) can be modelled either by the mean term or the noise term, it is unclear to me that how the authors implementation or design of the method can avoid this issue
        - probably some more careful study/experiments will be needed for clarify these issues.
        - while it is well expected, by doing this, the model can potentially learn well with much reduced resources cost, it seems to me the estimation problem will be less stable (more sensitive to hyperparameter choice), some relevant discussions on this regard will also be useful.
    - for empirical results at Table 4, the authors suggest a p-value<0.01, does this mean each p-value is calculated for these three datasets? It's probably better to list all these three p-values, especially the one for CIFAR100 experiment.
- novelty
    - the idea of reusing the mean term for noise term has been widely studied in statistics community, but might be the first time for this particular problem setting, so probably fine. In those studies, the identification issue is the biggest challenge, which leads to the question above.

**Strength And Weaknesses:**

- strength
    - the idea is straightfoward and a natural extension of the existing methods
    - the discussion and summary to previous methods are nice
- weakness
    - it seems unclear to me that where the strength of the methods come from
    - the empirical strength of the methods does not seem clear to me

**Summary Of The Paper:**

The paper studies heteroscedastic models and extends the previous studies that require an explicit modelling of the noise structure into a simple modelling by reusing the logits from the mean (non-noise part) term. And then the idea is further extended to contrastive learning setting.

**Summary Of The Review:**

reasonably good work, but some clarification on why the idea can bypass the identification issue might need to be deeply discussed.

---

> ### Author Response · Authors · 2022-11-15
> **Response to Reviewer FYog Part 1**
>
> We thank reviewer FYog for their helpful and constructive review.
>
> **"It seems unclear to me that where the strength of the methods come from."**
>
> - We would like to thank the reviewer for encouraging us to more formally ground our approach and to provide more explanations and insights as to why the method actually works.
> - First, we recall that HET also considers a low-rank covariance, say of rank R, with $\Sigma_\text{het}(\mathbf{x}) = \mathbf{Q}(\mathbf{x})^\top \mathbf{Q}(\mathbf{x})$ and $\mathbf{Q}(\mathbf{x})$ of shape (R, K). As a result, we stress that the goal of the parametrization of HET-XL is not to be as expressive as a full-rank K-by-K covariance but rather to be comparable to that of $\Sigma_\text{het}(\mathbf{x})$. For HET-XL, we have $\Sigma_\text{het-xl}(\mathbf{x}) = \mathbf{W}^\top \mathbf{P}(\mathbf{x})^\top \mathbf{P}(\mathbf{x}) \mathbf{W}$, with $\mathbf{W}$ of shape (D, K) and $\mathbf{P}(\mathbf{x})$ of shape (R, D), which is another rank-R parametrization. From this viewpoint, whether HET-XL can model a covariance similar to that of HET depends on the structures of the row spaces of $\mathbf{W}$ and $\mathbf{Q}(\mathbf{x})$.
> - In a new section in Appendix I.1, we quantitatively assess the alignment of these row spaces (via the smallest principal angle) in the case of HET for a ResNet152x1 on ImageNet-21k. We show that the row space of the $\mathbf{W}$ learned by HET naturally aligns with that of $\mathbf{Q}(\mathbf{x})$ (in expectation over the image $\mathbf{x}$ of the validation set). This implies that expressing the covariance through $\mathbf{W}$, as proposed by HET-XL, is a natural and well-motivated design choice.
>
> **Identifiability issue between mean and covariance term**
>
> - In HET-XL, the logits are modeled by $\mathbf{W}^\top \phi(\mathbf{x}) + \mathbf{W}^\top \epsilon(\mathbf{x})$, with  $\epsilon(\mathbf{x}) \sim \mathcal{N}(0, \Sigma(\mathbf{x}))$. We first would like to underline that the noise term has its mean equal to zero, which leaves the noise term with fewer degrees of freedom to possibly try to fit the logits. Second, as another constraint of our parametrization, we do not reuse the entire mean term ($\mathbf{W}^\top \phi(\mathbf{x})$) but just the matrix $\mathbf{W}$.  Further note that in Appendix I, we discuss precisely removing this weight sharing, doing so has little effect on the final results. This empirical result and our newly added discussion on the motivation for sharing $\mathbf{W}$, Appendix I.1, lead us to the HET-XL method with weight sharing enabled.
> - However, it is known from the Gaussian-process literature (e.g., see Sec. 5.4.1, Figure 5.5 of Rasmussen & Williams, 2006) that we may face identification issues when the noise term “absorbs” all the signal because of a bad local minimum of the marginal likelihood. We have never experienced any identification issues of that nature in our experiments, even when trying multiple initialisations of our model (we hypothesize this is due to the loss surface that may be easier to optimize than that of the marginal likelihood of a GP).  In terms of code, our implementation is as follows, without any specific logic to deal with possible identification issues:
> ```
> # B=batch size, D=pre-logits dimension, S=#MC samples
> # [B, D] -> [B, 1, D]
> prelogits = jnp.expand_dims(prelogits, axis=1)
> # [B, 1, D] -> [B, S, D]
> latents = prelogits + noise_samples
> # [B, S, D] -> [B, S, K]
> logits = self.pre_logits_to_logits_dense_layer(latents)
> ```
> - Finally, we would be grateful if the reviewer could give us literature pointers where the mean parameter is reused (_“...the idea of reusing the mean term for noise term has been widely studied in statistics community…”_), in particular in setups matching ours, where only the $\mathbf{W}$ matrix is shared.
>
> **"Estimation problem will be less stable (more sensitive to hyperparameter choice), some relevant discussions on this regard will also be useful."**
>
> The identifiability issue raised by the reviewer may not apply as we do not share the mean term of the Gaussian, merely the corresponding weight matrix. This may have a bearing on the reviewers comments on hyperparameter sensitivity. Nonetheless, we note that we reuse, without modification, *all* the hyperparameters of the HET method which have been tuned in prior work (Tran, et al. 2022). The only hyperparameters introduced by the HET-XL method are the boundaries of the learned temperature parameter which we set to be min=0.05 and max=5.0 for all experiments. We find that performance is insensitive to these boundaries provided the boundaries contain the final temperature learned by HET-XL, see appendix K. Therefore we conclude that HET-XL is relatively insensitive to hyperparameter choice, given that all existing hyperparameters can be reused and we remove the need for tuning the main sensitive hyperparameter from the HET method (the temperature).

---

> > ### Comment · Reviewer_FYog · 2022-11-15
> > **Response to rebuttal**
> >
> > Thanks for the clarification, most of these clarifications have addressed my concerns quite nicely, so I have updated my score to the positive end.
> >
> > However, I would recommend the authors include these detailed discussions such as "never experienced any identification issues of that nature in our experiments", "we conclude that HET-XL is relatively insensitive to hyperparameter choice", and the ones about p-values in the future version of the paper.

---

> > > ### Author Response · Authors · 2022-11-16
> > > **Thanks and quick update**
> > >
> > > We thank the reviewer for their quick response and updating their score. We have implemented their suggestion to include the discussion of the p-values and hyperparameter sensitivity in the main paper and have updated the paper accordingly, see Appendix A.2 and N. We will also happily include the discussion on identifiability issues.

---

> ### Author Response · Authors · 2022-11-15
> **Response to Reviewer FYog Part 2**
>
> **P-values of Table 4 (now Table 6 after revision)**
>
> We thank the reviewer for raising this clarifying question. To clarify the computation of the p-values: we compute 3 separate p-values using 3 separate two-sample unequal variance t-tests. We compute a p-value comparing the DET vs. HET-XL Imagenet zero-shot accuracy, a separate p-value for the CIFAR-100 results and similarly for Oxford-IIIT Pet. Only the ImageNet results are significant at the 0.01 level of significance. Note however that the other two computed p-values are well above this level of significance, hence we did not report the exact numbers in the paper. Nonetheless for clarity we report here the CIFAR-100 p-value: 0.839 and Oxford-IIIT Pet p-value: 0.596.

---

### Official Review · Reviewer_EUZN · 2022-10-24

**Confidence:** 3
**Correctness:** 4
**Technical Novelty And Significance:** 3
**Empirical Novelty And Significance:** 3
**Recommendation:** 8

**Clarity, Quality, Novelty And Reproducibility:**

The paper is easy to follow, extensive evaluation demonstrates the value of the contribution. Reproducibility of some experiments is limited due to use of proprietary datasets. However, authors also report result on publicly available data.

**Strength And Weaknesses:**

Strength:
-	Simplicity of the approach.
-	Extensive empirical study on large scale datasets.
-	Strong results that show both scalability of the approach and good performance.

Weaknesses:
- Lack of evaluation on downstream transfer tasks: Usually, representations learned from large-scale supervised datasets are then evaluated in a transfer setting through finetuning or linear probing. Would you expect HET-XL to show a gain in that setting, beyond the zero-shot case.
- HET extra cost due to Monte-Carlo Sampling: HET requires to perform a MC sampling to compute the network output, which increase the computational cost of the forward prop compared to a deterministic model. How would a deterministic model with the same computational budget for the forward prop than HEC perform?


**Summary Of The Paper:**

This paper investigates the use of heteroscedastic classifiers for large scale classification problems. Heteroscedastic classifiers consider uncertainty in the decision function by learning a multivariate gaussian distribution over classifier logits. This paper proposes two simple tricks to scale such approach forclassification problems with large number of classes:
 1) add gaussian noise at the pre-logit level of the networks, to remove the parameter dependencies on the number of classes
 2) learn the softmax temperature instead of more involve strategies to stabilize the MC  sampling processing.

Extensive experimentation on large scale datasets (ImageNet21K, JFT-300M/4B) shows the good scalability of their approaches and good performances over deterministic baseline and a strong hashing-based baseline.

Authors also apply their approach to a contrastive learning setting and investigate the 0-shot classification setup.


**Summary Of The Review:**

The paper proposes a simple approach with good scalability property and extensive evaluation to demonstrate the value of their proposal. I therefore support acceptance.

---

> ### Author Response · Authors · 2022-11-15
> **Response to Reviewer EUZN**
>
> We thank reviewer EUZN for their helpful and constructive review.
>
> **Lack of evaluation on downstream transfer tasks.**
>
> We are grateful for the reviewer’s suggestion to complete our evaluation on downstream transfer tasks. To that end, we have added a new section in Appendix O with few-shot results (with linear probing) over 9 datasets. In summary, Table 22 shows that HET-XL consistently improves upon DET for {5, 10, 25} shots while matching the performance of DET for the 1-shot evaluation.
>
> **Comparison with deterministic model with the same computational budget as that of HET**
>
> We will address this comment in a follow-up reply as soon as the relevant experiments have completed.

---

> ### Author Response · Authors · 2022-11-16
> **Response to Reviewer EUZN part 2**
>
> **Comparison with deterministic model with the same computational budget as that of HET**
>
> We thank the reviewer for this suggested experiment. We have followed the procedure of (Collier et al, 2022) who expanded the dimension of the final layer in the DET network to equalize the parameter count between the DET and HET model. In Appendix P, we increase the capacity of the DET model to equal that of the HET-XL model and observe that for the ViT-L/32 model trained on JFT-300M for 7 epochs, the DET model with increased capacity achieves Prec@1: 0.470 and NLL: 7.82. This represents a small improvement on the results in Table 3 which shows the lower capacity DET model with Prec@1: 0.468 and NLL: 7.83. Those results are still far behind the HET-XL model which achieves Prec@1: 0.498 and NLL: 7.65. Therefore, we conclude that HET-XL's performance gains cannot simply be attributed to increased model capacity.

---

### Official Review · Reviewer_mUAK · 2022-10-24

**Confidence:** 3
**Correctness:** 4
**Technical Novelty And Significance:** 2
**Empirical Novelty And Significance:** 3
**Recommendation:** 6

**Clarity, Quality, Novelty And Reproducibility:**

Clarity - While I think the paper could try to more closely tie the individual contributions together, overall the paper is clearly written and understandable.

Quality - There are no obvious issues in quality.

Novelty - As stated above, the novelty of this work is low due to the

Reproducibility - While the authors seem to intend to release their code on GitHub, some of their results are on proprietary data, making reproducibility impossible.

**Strength And Weaknesses:**

Strengths
1. Overall the method is presented in a clear manner.  It is both easy to see why the method would result in models with fewer parameters and likely easy to implement.
2. The empirical results seem to justify the changes to standard heteroscedastic learning as proposed in the paper.

Weaknesses
1. The biggest weakness is that the the work lacks novelty.  Each of the three contributions is simply applying well known techniques with slight variations to reduce practical run time of training and inference.  The paper provides no principled or theoretical contribution, which makes the paper seem like more incremental than substantial.
2. The empirical results are not very compelling.  First, the number of epochs that each model is trained for does not seem representative of complete training procedures.  The JFT-4B (which are from a proprietary data set, and thus is impossible to replicate) results are from training from a single epoch, for instance.  A more compelling result would be to plot the NLL and/or precision of the model as a function of wall run time, as this would be illustrate the claimed main benefit of HET-XL: With fewer parameters, HET-XL is able to scale better than HET methods.  Further, the ablation results need to be in the main paper and more clearly presented in the context of the results using the individual components.  The memory and train timing figures are a good idea in principle, but it is unclear how the number of MC samples practically effects training.

**Summary Of The Paper:**

The authors propose a method for performing Heteroscedastic classification that can scale independently of the number of classes.  The method simply defines noise not over the logits of a network as is commonly done, but over the layer previous to the logins.  In this way the learned instance-specific covariance of the noise distribution is a DxD matrix (where D is the dimensionality of the pre-logit representation) instead of KxK (where K is the number of classes).  The authors also provide some empirical justification for treating the commonly validated temperature Hyperparameter as a free parameter in the training procedure.  In addition to those two contributions, authors define a contrastive learning model that uses their proposed noise model.  These three contributions are aimed at scaling heteroscedastic classification to cases where there are a considerable (> 10,000) classes.  Empirically, their method out-performs both heteroscedastic and deterministic baselines in terms of negative log likelihood and precision while producing models with fewer parameters than other heteroscedastic baselines.

**Summary Of The Review:**

While I feel the contributions of this paper likely improve the practice of learning heteroscedastic classifiers, I feel they are incremental and not significant enough to warrant acceptance.

(See response to rebuttal for change of final score)

---

> ### Author Response · Authors · 2022-11-15
> **Response to Reviewer mUAK part 1**
>
> We thank reviewer mUAK for their helpful and constructive review.
>
> **"The paper provides no principled or theoretical contribution."**
>
> - We would like to thank the reviewer for encouraging us to more formally ground our approach and to provide more explanations and insights as to why the method actually works.
> - First, we recall that HET also considers a low-rank covariance, say of rank R, with $\Sigma_\text{het}(\mathbf{x}) = \mathbf{Q}(\mathbf{x})^\top \mathbf{Q}(\mathbf{x})$ and $\mathbf{Q}(\mathbf{x})$ of shape (R, K). As a result, we stress that the goal of the parametrization of HET-XL is not to be as expressive as a full-rank K-by-K covariance but rather to be comparable to that of $\Sigma_\text{het}(\mathbf{x})$. For HET-XL, we have $\Sigma_\text{het-xl}(\mathbf{x}) = \mathbf{W}^\top \mathbf{P}(\mathbf{x})^\top \mathbf{P}(\mathbf{x}) \mathbf{W}$, with $\mathbf{W}$ of shape (D, K) and $\mathbf{P}(\mathbf{x})$ of shape (R, D), which is another rank-R parametrization. From this viewpoint, whether HET-XL can model a covariance similar to that of HET depends on the structures of the row spaces of $\mathbf{W}$ and $\mathbf{Q}(\mathbf{x})$.
> - In a new section in Appendix I.1, we quantitatively assess the alignment of these row spaces (via the smallest principal angle) in the case of HET for a ResNet152x1 on ImageNet-21k. We show that the row space of the $\mathbf{W}$ learned by HET naturally aligns with that of $\mathbf{Q}(\mathbf{x})$ (in expectation over the image $\mathbf{x}$ of the validation set). This implies that expressing the covariance through $\mathbf{W}$, as proposed by HET-XL, is a natural and well-motivated design choice.
>
> **Efficiency of HET and HET-XL with respect to the MC samples: "it is unclear how the number of MC samples practically effects training."**
> - We have clarified and further experimentally measured how HET and HET-XL depend on the number of MC samples. We have added the corresponding material in a new section, in Appendix F.1.
> - In a nutshell, we have conducted a fine-grained time-complexity analysis for both HET and HET-XL, which results in a non-trivial trade-off between K (number of classes), D (the dimension of the pre-logits), R (the number of factors of the low-rank parameterization) and the number of MC samples. Our analysis recovers the observed phenomenon wherein HET-XL is more efficient than HET for a smaller number of MC samples (which, in practice, is the setup we are mostly interested in). Moreover, our analysis remarkably agrees with the actual training times and prediction times. For example, on ImageNet-21k with ResNet152x1, the training time for HET-X  is faster than HET up to 16 MC samples, with HET being faster from 64 MC samples on.
> - Importantly, HET-XL does not need many MC samples to perform well (see Table 12, on ImageNet-21k with ResNet152x1). For instance, HET-XL has better Prec@1 and NLL at 1 MC sample than HET at 1000 MC samples, despite also being substantially faster to train (52% reduction in training time). This is in addition to the parameter count and memory usage gains already highlighted in the paper.
>
> **"Each of the three contributions is simply applying well known techniques with slight variations to reduce practical run time of training and inference."**
>
> - Although previous work has looked into approximating large covariance matrices, (see e.g., (Fan et al, 2015), and references therein) our method proposes a simple and elegant idea to bypass this problem by approximating the covariance matrix in the pre-logits space and leveraging the linearity of the final projection layer. We further provide empirical evidence (see Appendix I.1) suggesting that this is a natural design choice. To the best of our knowledge, this idea has not been explored so far in the context of heteroscedastic classification.
> - Temperature learning has been applied in the context of contrastive learning before; however in our setup the temperature drives the bias-variance trade-off; see (Collier et al, 2021) and therefore it plays a different role. In particular, temperature tuning in contrastive learning has typically been justified on the basis of correcting for the fact that the contrastively learned embeddings are normalized. We justify the learning of the temperature on a more principled basis: Appendix G.2 recognizes the particular structure of the temperature hyperparameter, in that the temperature appears in the training and validation objective. This insight is to the best of our knowledge novel and widely applicable to the machine learning community.
> - Finally, as shown by (Tay et al., 2022) in “Scaling Laws vs Model Architectures: How does Inductive Bias Influence Scaling?”, it is extremely challenging to maintain the benefits of a method over increasingly large scales, which we achieve with HET-XL up to the scale of ViT-L/32 and JFT-4B.

---

> ### Author Response · Authors · 2022-11-15
> **Response to Reviewer mUAK part 2**
>
> **“the number of epochs that each model is trained for does not seem representative of complete training procedures. The JFT-4B (which are from a proprietary data set, and thus is impossible to replicate) results are from training from a single epoch, for instance.”**
>
> Please note that Table 10, Appendix D contains JFT-4B results trained for 2 and 3 epochs. All methods benefit from the increased training steps, but HET-XL still outperforms DET by a similar margin. We report single epoch results in the main paper, as the use of a single training epoch for JFT-4B is standard in the literature, for example see the Plex paper (Tran, et al. 2022). Note that even using a single training epoch on JFT-4B results in 1.89 times as many training steps as training for 7 epochs on JFT-300M and 3.49 times as many training steps as training for 90 epochs on ImageNet-21k.
>
> **“A more compelling result would be to plot the NLL and/or precision of the model as a function of wall run time, as this would be illustrate the claimed main benefit of HET-XL: With fewer parameters, HET-XL is able to scale better than HET methods.”**
>
> - We thank the reviewer for suggesting to further emphasize the benefits of HET-XL in terms of wall-run time. We have added a new section in Appendix F.1, Table 12, illustrating that HET-XL does indeed scale better than HET.
> - For instance, on ImageNet-21k with ResNet152x1, HET-XL has better Prec@1 and NLL at 1 MC sample than HET at 1000 MC samples despite also being substantially faster to train (a 52% reduction in training time measured in TPU training hours).
>
> **“Further, the ablation results need to be in the main paper and more clearly presented in the context of the results using the individual components.”**
>
> Following the recommendations of the reviewer, we have re-organised the experiments in Section 6 to surface more ablation studies. Tables 4 and 5 have been moved to the main paper, as the model and data scaling ablations were positively commented on by reviewer HrTZ.
>
> **Reproducibility**
>
> We would like to re-emphasize that our ImageNet-21k experiments are fully reproducible with open source code on GitHub, similarly the layer code (in JAX) is available for application to any other setting. The other reviewers appreciated the reproducibility of our paper, not only in terms of code but also in terms of level of details describing the experiments. Finally, it is worth noting that having a blend of public and proprietary datasets for the evaluation of a paper is not a blocker for adoption, e.g., the original ViT paper (Dosovitskiy et al., 2020) used both ImageNet-21k and JFT-300M.

---

> > ### Comment · Reviewer_mUAK · 2022-11-22
> > **Response to rebuttal**
> >
> > Thank you for your rebuttal.  I really do think the proposed changes and additions to the paper (as well as pointing out Table 10 in appendix D) address all my concerns in terms of empirical results.  I still feel that the methodological and theoretic contributions are minimal, but since the main contribution of the work are more about the practice of learning heteroscedastic classifiers, I am inclined to increase my score to a weak accept as I am more convinced there is practical utility to the work.

---

### Official Review · Reviewer_r78j · 2022-10-25

**Confidence:** 4
**Correctness:** 4
**Technical Novelty And Significance:** 3
**Empirical Novelty And Significance:** 3
**Recommendation:** 6

**Clarity, Quality, Novelty And Reproducibility:**

The writing is clear and easy to understand. It clearly identifies a problem and proposes a solution. The solution is validated properly.

**Strength And Weaknesses:**

Strengths
- The manuscript is well-written and easy to understand.
- The proposed idea is simple yet effective.
- The proposed method has advanced training efficiency and scalability of learning heteroscedastic classifiers for extreme classification tasks using a simple idea.
- Extensive experimental results demonstrate the effectiveness of the proposed method.

Weaknesses
- In my understanding, the proposed method takes benefits from breaking the full rank assumption of the covariance matrix (from $K$ to $D$). The authors provide empirical evidence for this, but it is still unclear why it works. Could you provide any theoretical analysis for this? or How big does $D$ work well? Specifically, does the proposed method effectively work on ViT models having small dimensionality (e.g., ViT-small and ViT-tiny)?
- I think applying the proposed method to contrastive learning is an interesting idea. However, the reported improvements are marginal. However, under the perspective of instance discrimination view, there exist very strong class-correlations; for example, CIFAR-100 has actual 100 categories but 50,000 classes in this case. I am curious about whether we can expect heteroscedastic classifiers to work well under such a strong class correlation. Moreover, in contrastive learning, I think adding noise before and after $l_2$ normalization would have different behaviors.


**Summary Of The Paper:**

This manuscript proposes HET-XL, which is a simple yet effective solution for handling heteroscedastic classifiers for extreme classification. Specifically, the proposed method has advanced the scaling issues in the existing work. The extensive experiments and ablation studies showed the effectiveness of the proposed method on large-scale image classification benchmarks such as JFT-300M and ImageNet-21k.

**Summary Of The Review:**

Overall, I would recommend the acceptance, as it has solid contributions and is worth sharing. Although the proposed solution is straightforward, it clearly advanced the previous works and has shown significant improvements on large-scale benchmarks.

---

> ### Author Response · Authors · 2022-11-15
> **Response to Reviewer r78j**
>
> We thank reviewer r78j for their helpful and constructive review.
>
> **"In my understanding, the proposed method takes benefits from breaking the full rank assumption of the covariance matrix (from K to D). The authors provide empirical evidence for this, but it is still unclear why it works. Could you provide any theoretical analysis for this?"**
>
> - We would like to thank the reviewer for encouraging us to more formally ground our approach and to provide more explanations and insights as to why the method actually works.
> - First, we recall that HET also considers a low-rank covariance, say of rank R, with $\Sigma_\text{het}(\mathbf{x}) = \mathbf{Q}(\mathbf{x})^\top \mathbf{Q}(\mathbf{x})$ and $\mathbf{Q}(\mathbf{x})$ of shape (R, K). As a result, we stress that the goal of the parametrization of HET-XL is not to be as expressive as a full-rank K-by-K covariance but rather to be comparable to that of $\Sigma_\text{het}(\mathbf{x})$. For HET-XL, we have $\Sigma_\text{het-xl}(\mathbf{x}) = \mathbf{W}^\top \mathbf{P}(\mathbf{x})^\top \mathbf{P}(\mathbf{x}) \mathbf{W}$, with $\mathbf{W}$ of shape (D, K) and $\mathbf{P}(\mathbf{x})$ of shape (R, D), which is another rank-R parametrization. From this viewpoint, whether HET-XL can model a covariance similar to that of HET depends on the structures of the row spaces of $\mathbf{W}$ and $\mathbf{Q}(\mathbf{x})$.
> - In a new section in Appendix I.1, we quantitatively assess the alignment of these row spaces (via the smallest principal angle) in the case of HET for a ResNet152x1 on ImageNet-21k. We show that the row space of the $\mathbf{W}$ learned by HET naturally aligns with that of $\mathbf{Q}(\mathbf{x})$ (in expectation over the image $\mathbf{x}$ of the validation set). This implies that expressing the covariance through $\mathbf{W}$, as proposed by HET-XL, is a natural and well-motivated design choice.
>
> **How big does D work well? Specifically, does the proposed method effectively work on ViT models having small dimensionality (e.g., ViT-small and ViT-tiny)?**
>
> We thank the reviewer for suggesting this interesting experiment. In Table 16, Appendix I, we have added an ablation along these lines. We remove the sharing of the $\mathbf{W}$ matrix, therefore enabling $\epsilon’(\mathbf{x})$ to be of different dimensionality to D, referred to as M. We then vary the dimension of $\epsilon’(\mathbf{x})$ and observe the effect on the performance. We can see that indeed, as we increase M to D, as is done in HET-XL, we see improved performance. Note however that we can decrease M by a factor of 8 from D=2048 to M=256, and still have better performance than HET. HET-XL’s performance is better than DET’s even at M=64.
>
> **"CIFAR-100 has actual 100 categories but 50,000 classes in this case. I am curious about whether we can expect heteroscedastic classifiers to work well under such a strong class correlation."**
>
> We would like to clarify that the evaluation for CIFAR-100 corresponds to zero-shot classification, so that we are not subject to the concerns raised by the reviewer. The contrastive learning happens over the same dataset as that used by Zhai et al., 2022.
>
> **"Adding noise before and after l2 normalization would have different behaviors."**
>
> We thank the reviewer for suggesting this ablation study. In Appendix A.2, Table 7, we have added the results when (i) normalizing before adding the noise, (ii) normalizing after, as well as (iii) normalizing before and after. In summary, we observe it is especially important to normalize before adding the noise, while a re-normalization after has a mild influence. The results of our core paper use (i).

---

### Official Review · Reviewer_HrTZ · 2022-10-26

**Confidence:** 3
**Correctness:** 4
**Technical Novelty And Significance:** 2
**Empirical Novelty And Significance:** 4
**Recommendation:** 8

**Clarity, Quality, Novelty And Reproducibility:**

**Clarity.**  The paper was very clearly written.  This is certainly above average in terms of writing quality, grammar, etc.

**Quality.**  The quality is also high, in the sense that the experiments seemed to have been conducted in a principled and thorough way.  The description of the architecture was also crisp.

**Novelty.**  There are aspects of this paper that are novel.  The empirical results in Table 3 show that HET-XL surpasses the state-of-the-art.  The method involves some nice tricks, which are novel in this problem setting.  However, overall this paper more or less uses standard or existing tools/tricks, so from the methodological side, it's not particularly novel in my opinion.

**Reproducibility.**  Clearly this would take a massive amount of compute to reproduce.  But controlling for that, it seems that the authors will release their code, and the descriptions given in the paper seem sufficient to be able to reproduce this work.

**Strength And Weaknesses:**

### Strengths

**Writing quality.**  This is the best-written paper in my batch of ICLR papers this year.  The arguments are clearly articulated and it's easy to understand the main ideas.

**Simplicity.**  I found the simplicity of the main ideas to be appealing.  The trick used to change the layer at which the noise is added is simple, and it's encouraging (as shown in an appendix) that trick comes at no loss of performance.  The main ideas seem like they would easy to implement for those who wanted to reproduce this work, and authors also seem committed to making their code available, which is a positive and encouraging sign.

**Experiments.**  The experiments are thorough and they show that the method seems to offer a performance improvement over both deterministic classifiers (henceforth DET) and heteroscedastic classifiers (henceforth DET).  Furthermore, relative to HET, the proposed architecture HET-XL has a smaller number of parameters; indeed, the authors show their architecture's parameter count does not scale with the number of classes $K$.  These two results -- both highlighted in Table 3 -- are exactly what one would hope to see.

Overall, the rest of the experiments also seem to be quite thorough.  In the main text, we get a couple of ablations concerning the latency and memory usage of the relevant algorithms.  There are also quite a few extra results in the appendix, including ablation studies on learning $\tau$, model scaling, data scaling, and the number of floating point operations.  In eacfh case, it seems that HET-XL compares favorably to the baselines.

## Weaknesses

**Minor points of confusion.**  Here are a couple of things that confused me while I was reading the paper:

* I don't understand why this model requires *marginalizing* over the noise distribution.  Granted, I'm not an expert on heteroscedastic classifiers, so perhaps I have missed something obvious here.  However, from my understanding, the noise is modelled by a Gaussian, which we assume has zero mean.  Furthermore, the covariance matrix of this Gaussian is learned from data.  One one has access to this covariance matrix, why does one need to run MCMC to sample from this noise distribution?  Can one not directly sample, given that the distribution is normal?  Perhaps it has something to do with needed to differentiate through the covariance estimation?  If the authors could elaborate on this point, I think it would make the paper clearer.

* What is the so-called "hashing trick?"  It's relatively difficult to understand the innovation in Section 6.1 without knowing this, and I think that it's possible that many readers of this paper may not have heard of it before.  In my opinion, it would be worth adding a few lines to the paper to describe the main idea behind this trick, and why it's useful here.

**Contrastive learning.**  Having read the paper, I'm not sure what the contribution of the contrastive learning section is.  Indeed, it seems clear that contrastive learning can be formulated as a massive classification problem.  However, the question is: Does this perspective result in better algorithms or architectures for contrastive learning.  And based on the results in Table 4, it's not clear that it does.  The table indicates that on ImageNet, the heteroscadastic classification view of contrastive learning yields a marginal (albeit, statistically significant) improvement.  However, on the other two datasets, HET-XL performs worse than the baseline deterministic classifier.  Therefore, it's somewhat unclear whether it's worth using a classifier with potentially many more parameters relative to DET in the setting of contrastive learning.  For these reasons, I do not see the contrastive results as an impactful contribution.  If I have missed something, I would welcome the thoughts of the authors on this point.

**Efficiency.**  After reading the paper, it seemed unintuitive that when taking many MC steps, HET-XL underperformed relative to HET in terms of latency.  Could the authors shed more light on this?  Why is sampling less efficient for HET-XL?  And if this is the case, in terms of time-complexity, would one then tend to prefer HET over HET-XL.  (Of course, in space-complexity, it does seem that HET-XL is to be preferred in these large-scale settings.)

**Summary Of The Paper:**

As indicated by the title, this paper considers the problem of scaling heteroscedastic classifiers to problems where the number of classes is large.  The main contribution is a simple trick which eliminates the dependence on the number of classes on the total number of parameters.  The authors also show that the temperature parameter commonly used for these networks does not need to be treated as a hyperparameter; rather, it can be optimized as a trainable parameter.  The authors discuss how these insights can be applied to contrastive learning, and then they provide thorough experiments showing that their method scales more efficiently than existing heteroschedastic classifies and that it outperforms a deterministic baseline.

**Summary Of The Review:**

Overall, I thought this was a solid paper.  The empirical results are impressive for the heteroscedastic setting.  The method is simple and clearly presented.  And the experiments are thorough.  On the negative side, aside from a few minor points of confusion, I would argue that the contrastive results are not particularly impactful and that the time-complexity of HET-XL is concerning.  I think that this paper has some empirical novelty, but it does not introduce tools that may be more broadly applicable to the learning community.  All of this being said, I think that this paper does exactly what it sets out to do, which is to improve the SOTA in heteroscedastic classification.  Therefore, I recommend that this paper be accepted.

---

> ### Author Response · Authors · 2022-11-15
> **Response to Reviewer HrTZ**
>
> We thank reviewer HrTZ for their helpful and constructive review.
>
> **Monte Carlo vs. Markov-chain Monte Carlo**
>
> We would like to clarify that none of the methods presented in the paper requires Markov chain Monte Carlo (MCMC) sampling. Instead, both HET and HET-XL use Monte Carlo (MC) sampling to approximate the expectation of Eq. (2) and (5), by simply taking samples from the Gaussian distributions of $\epsilon(\mathbf{x})$ and $\epsilon’(\mathbf{x})$. To help clarify the origin of this expectation, we have added a new section in Appendix A.1 (this new section is linked in Section 2.1). Since several reviewers asked about the computational impact of the MC sampling, we have also added another section in Appendix F.1 with a detailed analysis comparing HET and HET-XL (see other dedicated reply).
>
> **Details about the "hashing trick"**
>
> We thank the reviewer for suggesting to add more background about the "hashing trick", in order to better appreciate the method HET-H. We have accordingly added a new section in Appendix A.3 (linked from Section 6.1).
>
> **Contrastive results not an impactful contribution**
>
> HET-XL outperforms DET for both ImageNet and CIFAR-100, so only on Oxford-IIIT Pet is DET better (only the ImageNet result is statistically significant). Further note that we have added the following result to the paper in order to contextualize the magnitude of the contrastive learning gains: when the same DET model is trained for 4 billion rather than 18 billion examples, the zero-shot accuracy (average over 3 random seeds) is 84.98/83.14/97.57 for ImageNet, CIFAR-100 and Oxford-IIIT Pet respectively. Thus, the gains we see from HET-XL on ImageNet are similar in magnitude to increasing the number of training examples shown to DET by a factor of 4.5.
>
> **Efficiency of HET and HET-XL with respect to the MC samples**
>
> - We have clarified and further experimentally measured how HET and HET-XL depend on the number of MC samples. We have added the corresponding material in a new section, in Appendix F.1.
> - In a nutshell, we have conducted a fine-grained time-complexity analysis for both HET and HET-XL, which results in a non-trivial trade-off between K (number of classes), D (the dimension of the pre-logits), R (the number of factors of the low-rank parameterization) and the number of MC samples. Our analysis recovers the observed phenomenon wherein HET-XL is more efficient than HET for a smaller number of MC samples (which, in practice, is the setup we are mostly interested in). Moreover, our analysis remarkably agrees with the actual training times and prediction times. For example, on ImageNet-21k with ResNet152x1, the training time for HET-X  is faster than HET up to 16 MC samples, with HET being faster from 64 MC samples on.
> - Importantly, HET-XL does not need many MC samples to perform well (see Table 12, on ImageNet-21k with ResNet152x1). For instance, HET-XL has better Prec@1 and NLL at 1 MC sample than HET at 1000 MC samples, despite also being substantially faster to train (52% reduction in training time). This is in addition to the parameter count and memory usage gains already highlighted in the paper.
>
> **"Introduced tools may not be more broadly applicable to the learning community"**
>
> - Although previous work has looked into approximating large covariance matrices, (see e.g., (Fan et al, 2015), and references therein) our method proposes a simple and elegant idea to bypass this problem by approximating the covariance matrix in the pre-logits space and leveraging the linearity of the final projection layer. We further provide empirical evidence (see Appendix I.1) suggesting that this is a natural design choice. To the best of our knowledge, this idea has not been explored so far in the context of heteroscedastic classification and could be exploited more broadly in the machine learning community.
> - Temperature learning has been applied in the context of contrastive learning before; however in our setup the temperature drives the bias-variance trade-off (see Collier et al, 2021) and therefore it plays a different role. In particular, temperature tuning in contrastive learning has typically been justified on the basis of correcting for the fact that the contrastively learned embeddings are normalized. We justify the learning of the temperature on a more principled basis: Appendix G.2 recognizes the particular structure of the temperature hyperparameter, in that the temperature appears in the training and validation objective. This insight is to the best of our knowledge novel and widely applicable to the machine learning community.

---

> > ### Comment · Reviewer_HrTZ · 2022-11-18
> > **Rebuttal response**
> >
> > Thanks for your concise rebuttal.  I think that the additional background material on the hashing trick and the approximation of the expectation via sampling will clarify the paper for future readers.  The context for the contrastive experiments does help clarify the impact of using HET-XL vis-a-vis the deterministic classifier.  Overall, I think -- as I stated in my original review -- that this is a solid paper, and my score will remain at an 8.

---

### Official Review · Reviewer_GaFB · 2022-11-04

**Confidence:** 4
**Correctness:** 3
**Technical Novelty And Significance:** 2
**Empirical Novelty And Significance:** 4
**Recommendation:** 6

**Clarity, Quality, Novelty And Reproducibility:**

The paper is clear and benefits form the simplicity of the core idea.
The core idea appears simple and not particularly novel, but it is useful and interesting and appears to work surprisingly well.

**Strength And Weaknesses:**

Strengths:
- this is a simple technique to model heteroschedastic label noise and makes it tractable to large dimensions.
- the technique is enormously simple to utilize and plug in to existing classifiers.

Weakness:
- in theory, the expressivity of this parametrization should be lower than in the "full rank" scenario. However, the authors show graceful performance in practice. I feel it is noteworthy to keep this in mind in practice.

**Summary Of The Paper:**

The authors propose a technique for heteroschedastic classification, HET-XL, which does a simple trick to maintain scalability to settings with large amounts of classes.

The key idea is deceptively simple: the authors elect to model the covariance over noise not over class-space in the logits, but over the empedding space pre-logits, leading to a covariance matrix of much smaller dimension int he large class scenario which maintains tractability at ease.

Experiments show this approach also confers some regularization benefits and remains indeed tractable.

**Summary Of The Review:**

The authors propose a simple trick to make heteroschedastic classification scalable to large class-scenarios.
Their key idea is simple, easily applicable, and although it appears to lose out on some modeling oompf theoretically performs quite well in practice.

I think this could be useful for practitioners going after raw performance, though I would caution not to overinterpret the learned covariance matrix semantics as equivalent to one learned in class-space.

---

> ### Author Response · Authors · 2022-11-15
> **Response to Reviewer GaFB**
>
> We thank reviewer GaFB for their helpful and constructive review.
>
> **"In theory, the expressivity of this parametrization should be lower than in the "full rank" scenario. However, the authors show graceful performance in practice. I feel it is noteworthy to keep this in mind in practice."**
>
> - We would like to thank the reviewer for encouraging us to more formally ground our approach and to provide more explanations and insights as to why the method actually works.
> - First, we recall that HET also considers a low-rank covariance, say of rank R, with $\Sigma_\text{het}(\mathbf{x}) = \mathbf{Q}(\mathbf{x})^\top \mathbf{Q}(\mathbf{x})$ and $\mathbf{Q}(\mathbf{x})$ of shape (R, K). As a result, we stress that the goal of the parametrization of HET-XL is not to be as expressive as a full-rank K-by-K covariance but rather to be comparable to that of $\Sigma_\text{het}(\mathbf{x})$. For HET-XL, we have $\Sigma_\text{het-xl}(\mathbf{x}) = \mathbf{W}^\top \mathbf{P}(\mathbf{x})^\top \mathbf{P}(\mathbf{x}) \mathbf{W}$, with $\mathbf{W}$ of shape (D, K) and $\mathbf{P}(\mathbf{x})$ of shape (R, D), which is another rank-R parametrization. From this viewpoint, whether HET-XL can model a covariance similar to that of HET depends on the structures of the row spaces of $\mathbf{W}$ and $\mathbf{Q}(\mathbf{x})$.
> - In a new section in Appendix I.1, we quantitatively assess the alignment of these row spaces (via the smallest principal angle) in the case of HET for a ResNet152x1 on ImageNet-21k. We show that the row space of the $\mathbf{W}$ learned by HET naturally aligns with that of $\mathbf{Q}(\mathbf{x})$ (in expectation over the image $\mathbf{x}$ of the validation set). This implies that expressing the covariance through $\mathbf{W}$, as proposed by HET-XL, is a natural and well-motivated design choice.
>
> **Interpretation of the learned covariance matrix semantics?**
>
> We thank the reviewer for this proposition. It is a nice idea to explore whether the parameterization of HET-XL still preserves the interpretability that the covariance learned by HET offers. We are working on an analysis along the lines of Section 4.1 in Collier et al., 2021 where we will focus on ImageNet-21k instead of ImageNet-1k. We plan to add the results of that analysis either in the revised manuscript, if finished by the end of the review period, or in the camera ready.

---

> > ### Comment · Reviewer_GaFB · 2022-11-22
> > **Thank you for your response**
> >
> > Thank you, I appreciate these additions to your solid paper.

---

### Author Response · Authors · 2022-11-15
**Collective Response to Reviewers**

We thank the reviewers for their useful comments and reviews. We particularly appreciate that the reviewers found the method simple (GaFB/HrTZ/r78j/mUAK/EUZN/FYog) but effective with strong results _“Extensive experimental results demonstrate the effectiveness of the proposed method”_, _“extensive evaluation demonstrates the value of the contribution”_, (GaFB/HrTZ/r78j/mUAK/EUZN), clearly presented _“This is the best-written paper in my batch of ICLR papers this year.”_ (GaFB/HrTZ/r78j/mUAK/EUZN/FYog) and the availability of code to reproduce the results was noted (HrTZ).

We have responded to each reviewer individually, however we quickly summarize our response here.

Any changes with respect to our original submission is highlighted in purple in the updated text.

- Suggested by reviewer EUZN, we have added few-shot evaluations of the representations learned by HET-XL in Table 22, Appendix O.
- We have added a section, Appendix I.1, assessing the alignment of the row spaces of $\mathbf{W}$ and $\mathbf{Q}(\mathbf{x})$ in response to comments from reviewers (GaFB/r78j/mUAK/FYog) asking about the expressivity, theoretical grounding and why HET-XL works.
- In Appendix F.1, in response to reviewers (HrTZ/mUAK), we have added a time-complexity analysis and new results varying the MC samples for the HET baseline, enabling a more fine-grained study of the trade-off between computational cost and performance for HET and HET-XL.
- Following the suggestion of reviewer r78j, we have added new results, Table 16 in Appendix I, varying the dimensionality of $\epsilon’(\mathbf{x})$.
- We have ablated the positioning of the normalization layers in the contrastive learning setup, Appendix A.2, Table 7, in response to reviewer r78j.
- We have added new explanatory sections on MC sampling and the “hashing trick” in response to reviewer HrTZ.
- We have further clarified the significance of the contrastive learning gains for reviewer HrTZ by adding new DET baseline numbers.
- Based on the suggestion of reviewer mUAK, we have moved the model and data scaling ablations from the Appendix to the main paper.
- We have clarified multiple questions and/or misunderstandings raised by the reviewers about the existing content of the paper.

---

### Decision · Program_Chairs · 2023-01-20

**Decision:**

Accept: poster

**Justification For Why Not Higher Score:**

Due to the simple nature of the algorithmic improvement, a broader consideration of tasks and benchmarks (as well as possibly a theoretical analysis) could be performed to ascertain when this parametrization doesn't cause a drop in performance.

**Justification For Why Not Lower Score:**

The improved performance across the benchmarks considered, and with such simple (thus, easy to implement) algorithmic changes is a strong selling point for the paper.

**Metareview: Summary, Strengths And Weaknesses:**

The paper considers the problem of scaling models designed to function in settings when the number of classes K is very large, and one wants to explicitly model noise in the prediction layer. The obvious way to do this would be to have a K x K covariance matrix for some noise to be added after the softmax layer --- which would be prohibitive for large K. Restricting the rank of this matrix to be small would drop this cost to ~R x K, however this seems to come with performance drops in practice.

The paper introduces two very simple tricks that seem to come at no performance loss, and much improved scaling with K: simply add Gaussian noise on the penultimate layer in the network (of dimension D) prior to applying the softmax. This matrix requires only D^2 terms to parametrize, which can be much smaller for D small (which is the case for many popular architectures). Another simple trick the authors provide is to treat the temperature in the softmax as just another trainable parameter (rather than a hyperparameter).

Overall, the reviewers liked that the algorithmic change is so small, yet has good performance across the settings considered. During the discussion phase, the reviewers also asked for several extra experiments (e.g. slight variations in settings, p values for some of the experiments, some heuristic explanations for when the method might work or not) --- which the authors provided, and this improved the paper as well.

**Note From Pc:**

if the above contains the word "oral" or "spotlight" please see: "oral" presentation means -> notable-top-5% and "spotlight" means -> notable-top-25%. As stated in our emails, we are disassociating presentation type from AC recommendations